# Mosquito-Borne Diseases and Their Control Strategies: An Overview Focused on Green Synthesized Plant-Based Metallic Nanoparticles

**DOI:** 10.3390/insects14030221

**Published:** 2023-02-23

**Authors:** Hudson Onen, Miryam M. Luzala, Stephen Kigozi, Rebecca M. Sikumbili, Claude-Josué K. Muanga, Eunice N. Zola, Sébastien N. Wendji, Aristote B. Buya, Aiste Balciunaitiene, Jonas Viškelis, Martha A. Kaddumukasa, Patrick B. Memvanga

**Affiliations:** 1Department of Entomology, Uganda Virus Research Institute, Plot 51/59 Nakiwogo Road, Entebbe P.O. Box 49, Uganda; 2Laboratory of Pharmaceutics and Phytopharmaceutical Drug Development, Faculty of Pharmaceutical Sciences, University of Kinshasa, Kinshasa B.P. 212, Democratic Republic of the Congo; 3Centre de Recherche et d’Innovation Technologique en Environnement et en Sciences de la Santé (CRITESS), University of Kinshasa, Kinshasa B.P. 212, Democratic Republic of the Congo; 4Department of Biological Sciences, Faculty of Science, Kyambogo University, Kampala P.O. Box 1, Uganda; 5Department of Chemistry, Faculty of Science, University of Kinshasa, Kinshasa B.P. 190, Democratic Republic of the Congo; 6Institute of Horticulture, Lithuanian Research Centre for Agriculture and Forestry, 54333 Babtai, Lithuania

**Keywords:** mosquito-borne diseases, control strategies, green metallic nanoparticles, repellent properties, mosquitocidal activities

## Abstract

**Simple Summary:**

Mosquitoes are the carrier of pathogens that cause common human diseases such as malaria, Dengue, Chikungunya, yellow fever, Zika, and West Nile. The use of chemical and biological insecticides is among the key control tools for reducing mosquito populations at different life stages (i.e., eggs, larvae, pupae, and adults). The weaknesses of these tools, especially those with chemicals, are the high cost of production, and their negative effects on other beneficial organisms such as bees and water-dwelling invasive species. Thus, researchers and academics are searching for new safe and environmentally friendly forms of insecticides. Of late, green synthesized plant-based metallic nanoparticles have attracted great interest as an alternative to traditional insecticides. Typically using metal salts in combination with aqueous plant extracts, the synthesis of these nanoparticles is eco-friendly and relatively cheap. This review aims to report on the currently available knowledge on the development of green synthesized metallic nanoparticles and their performance as repellent, ovicidal, larvicidal, pupicidal, and adulticidal agents against different mosquito species.

**Abstract:**

Mosquitoes act as vectors of pathogens that cause most life-threatening diseases, such as malaria, Dengue, Chikungunya, Yellow fever, Zika, West Nile, *Lymphatic filariasis*, etc. To reduce the transmission of these mosquito-borne diseases in humans, several chemical, biological, mechanical, and pharmaceutical methods of control are used. However, these different strategies are facing important and timely challenges that include the rapid spread of highly invasive mosquitoes worldwide, the development of resistance in several mosquito species, and the recent outbreaks of novel arthropod-borne viruses (e.g., Dengue, Rift Valley fever, tick-borne encephalitis, West Nile, yellow fever, etc.). Therefore, the development of novel and effective methods of control is urgently needed to manage mosquito vectors. Adapting the principles of nanobiotechnology to mosquito vector control is one of the current approaches. As a single-step, eco-friendly, and biodegradable method that does not require the use of toxic chemicals, the green synthesis of nanoparticles using active toxic agents from plant extracts available since ancient times exhibits antagonistic responses and broad-spectrum target-specific activities against different species of vector mosquitoes. In this article, the current state of knowledge on the different mosquito control strategies in general, and on repellent and mosquitocidal plant-mediated synthesis of nanoparticles in particular, has been reviewed. By doing so, this review may open new doors for research on mosquito-borne diseases.

## 1. Introduction

Increased globalization, climate change, and human mobility have led to the ecological expansion of highly invasive species [1]. These invasive organisms, among which are arthropods, cause diseases that are deadly and lead to epidemics or pandemics [2]. Most important in that regard are mosquitoes (Diptera: Culicidae) that act as vectors of a variety of harmful pathogens and parasites [3,4]. Of these, the genera *Anopheles*, *Aedes*, and *Culex* are the most problematic vectors of most important pathogens, causing diseases such as malaria, Dengue, yellow fever, filariasis, Japanese encephalitis, and Zika [4,5]. There are various approaches being used to control mosquito-borne diseases and these include the use of several behavioral, chemical, biological, and mechanical methods. Various success levels have been achieved but these have been limited by a lack of effective vaccines and delays in the development of antiviral drugs for most arboviruses. The increasing insecticide resistance in the mosquito vectors, hampering the development of new drugs and vaccines, curtails these efforts. The failing vector control strategies, the proliferation of invasive mosquitoes as well as increased contact between humans and these vectors have led to the constant re-emergence of arboviruses [6]. Mosquito control programs are therefore faced with significant and rapidly changing challenges that necessitate the development of new approaches in the detection and control of diseases as a new requirement in public health.

Through the use of nano(bio)technology, which is gradually becoming recognized as a future emerging technology due to exceptional new benefits [7], nanoparticle development and technology could be used in the control of vectors and disease outbreaks. This could be accomplished through (a) the development of new drugs with increased activity, decreased toxicity, and sustained release; (b) green synthesis of new repellent formulations based on natural or synthetic compounds; (c) vector control through the use of green nanoparticles with repellent, adulticidal, or larvicidal activities [8]; and (d) the development of biosensors capable of rapidly detecting and diagnosing mosquito-borne viral diseases [9,10,11]. When used in the development of drug-related products, green synthesis of metallic nanoparticles would be an inexpensive and one-step process that would not require high pressure, energy, temperature, or highly toxic chemicals. The development of nanomedicines is promising new treatments for different mosquito-borne diseases, improved efficacy and bioavailability of products and drugs, with controlled release formulations that require optimal doses and consequently fewer adverse effects.

The objective of this review is to present the different mosquito vector control approaches with a focus on greenly synthesized nanoparticles, discuss the strengths and weaknesses of these different control strategies as well as provide recommendations for exploring new perspectives for the control of invasive mosquito species by green nanotechnology.

## 2. Mosquito-Borne Diseases

Mosquito-borne diseases are spread by the bite of infected female mosquitoes. The main mosquito-borne diseases include malaria, Chikungunya, Zika, Dengue, West Nile, yellow fever, Rift Valley fever, *Lymphatic filariasis*, and tick-borne encephalitis [12]. Important vectors of the pathogens causing these diseases are mosquitoes belonging to the genera *Aedes*, *Culex*, and *Anopheles*, widely distributed in Africa, Asia, South America, and Europe [13,14]. Approximately seven hundred million people around the world suffer from mosquito-borne illnesses, resulting in over one million deaths [15].

Of these, malaria, which is transmitted to humans by infected female *Anopheles* mosquitoes, poses the most life-threatening conditions [14,16]. The malaria burden spans from Africa’s tropics to Asia and South America [16,17]. In 2020, an estimated 241 million cases of malaria were reported worldwide, with 627 thousand people dying, the majority of whom were children in Africa [18,19].

West Nile is a mosquito-borne disease caused by the West Nile virus which is an enveloped, positive-strand ribonucleic acid (RNA) flavivirus of the family Flaviviridae [20]. The virus is primarily responsible for bird infections, especially in crows and blue jays, but it can also infect humans, dogs, horses, and other animals [21]. The mosquito genus *Culex*, especially species such as *Cx. quinquefasciatus*, *Cx. stigmatosoma*, *Cx. thriambus*, *Cx. pipiens*, and *Cx. nigripalpus*, affecting a range of avian and mammalian species are the main vectors responsible for the transmission of West Nile virus [22]. However, the virus can also be transmitted by several species of birds [23].

Dengue (fever) is caused by the Dengue virus, a positive (+)-stranded ribonucleic acid (RNA) virus that also belongs to the genus Flavivirus, and family Flaviviridae [24]. The virus is mainly transmitted through the bite of an infected female *Aedes aegypti* and *Ae. albopictus*, and zoonotic agents with no known arthropod vector [25]. The disease is the leading cause of morbidity and mortality throughout the tropics and sub-tropics, with an estimated 10,000 deaths and 100 million symptomatic infections per year in over 125 countries [26,27].

Yellow fever disease is caused by the yellow fever virus. This flavivirus is endemic in tropical Africa, South America, and North America. Yellow fever can cause extensive epizootics in non-human primates and outbreaks of human cases [28]. Pathogens causing the disease are mainly transmitted in Africa by *Ae. albopictus* and *Ae. aegypti* [29]. A component of the sylvatic reservoir system is in the non-human primates [29]. The disease transmission involves urban (human and urban mosquitoes) and sylvatic (non-human primates and forest-dwelling mosquitoes) cycles [30,31]. Sylvatic cycles account for the majority of the reported cases in South America [32]. By contrast, the intermediate cycle, which involves the transmission of yellow fever virus from monkey to human or from human to human through mosquito bites, is currently only reported in Africa [33,34].

Chikungunya is caused by an arthropod-borne alphavirus, Chikungunya virus [35]. This virus is primarily transmitted through the bite of infected *Aedes* (subgenus *Stegomyia*) mosquitoes (e.g., *Ae. aegypti*, *Ae. albopictus*, *Ae. Furcifer*, and other members of the *Ae. furcifer-taylori* group) [36]. Chikungunya virus is endemic in tropical and subtropical Africa and Southeast Asia, where transmission cycles between non-human primates and *Aedes* spp. exist [36].

Infection with the Rift Valley fever (RVF) virus (Phlebovirus, family Bunyaviridae) is a disease of domestic livestock such as cattle, sheep, and goats, as well as humans [37]. The disease is transmitted by mosquitoes or direct contact with infected animals and their products [38]. Mosquito species, such as *Ae. vexans* (Meigen), *Cx. poicilipes* (Theobald), and *Cx. quinquefasciatus*, are the main vectors responsible for the transmission of RVF [39]. Rift Valley fever is also transmitted by some species of the genus *Mansonia* and *Anopheles* [39]. The prevalence of RVF is highest in sub-Saharan Africa but recently the disease has been reported in Arabian Peninsulas and Saudi Arabia [40,41,42].

The Zika virus disease is caused by a flavivirus [43]. Since 2007, outbreaks of Zika virus have been reported in Micronesia and Brazil [44]. Moreover, from 2015 to 2017, approximately 6000 cases of symptomatic Zika virus disease were reported in the United States of America [45]. Like other flaviviruses, the Zika virus is spread by mosquitoes, primarily the *Aedes* (*Stegomyia*) genus, and species such as *Ae. aegypti*, *Ae. africanus*, *Ae. hensilli*, and *Ae. albopictus* have been implicated [46]. Although viral isolation studies suggest that *Ae. albopticus* was the likely vector in a 2007 Zika virus outbreak in Gabon, the dominant *Aedes* species vector has not been definitively identified in Africa [47]. Other modes of Zika virus transmission include maternal–fetal transmission, sexual transmission, blood transfusions, organ transplantation, and laboratory exposure [48].

*Lymphatic filariasis* (LF) is caused by microfilaria of *Wuchereria bancrofti*, *Brugia malayi*, or *Brugia timori* [49]. Depending on the geographical distribution, different species of the following mosquito genera are vectors of *Lymphatic filariasis*: (i) *Culex (Cx. annulirostris*, *Cx. bitaeniorhynchus*, *Cx. quinquefasciatus*, and *Cx. pipiens*); (ii) *Anopheles* (*An. arabinensis*, *An. bancroftii*, *An. farauti*, *An. funestus*, *An. gambiae*, *An. koliensis*, *An. melas*, *An. merus*, *An. punctulatus*, and *An. wellcomei*); (iii) *Aedes* (*Ae. aegypti*, *Ae. aquasalis*, *Ae. bellator*, *Ae. cooki*, *Ae. darlingi*, *Ae. kochi*, *Ae. polynesiensis*, *Ae. pseudoscutellaris*, *Ae. rotumae*, *Ae. scapularis*, and *Ae. vigilax*); (iv) *Mansonia* (*M. pseudotitillans* and *M. uniformis*); and (v) *Coquillettidia* (*C. juxtamansonia*) [50,51]. Globally, over 120 million people are affected by *Lymphatic filariasis* [52]. The disease is endemic in 73 countries, and 1.1 billion people in Asia, Africa, western Pacific, and parts of South America and the Caribbean are at risk of exposure to and contracting the disease [52].

Tick-borne encephalitis (TBE) is one of the most dangerous infections that can occur in humans. Tick-borne encephalitis virus (TBEV), the causative agent of TBE, is a member of the virus genus Flavivirus and the Flaviviridae family [53]. Transmitted by the bite of an infected arthropod, specifically mosquitoes and ticks, TBE involves the central nervous system [54]. This disease is very common in Central, Northern, and Eastern European as well as Asian countries. The highest notification rates of TBE in 2020 occurred in Lithuania, Latvia, Slovenia, and the Czech Republic, with incidences of up to 24 cases/100,000 individuals. In these endemic areas, the virus mortality reaches 0.5 to 4% [53,54].

## 3. Control Strategies for Mosquito-Borne Diseases

Various chemical, biological, and mechanical methods are used to control mosquito-borne diseases hence reducing their burden [55]. Some of them are briefly discussed below.

### 3.1. Chemical Control Strategies

The chemical methods commonly used for controlling mosquito-borne diseases include long-lasting insecticide-treated nets (LLINs), indoor residual spraying (IRS), peridomestic space spraying, and the use of mosquito repellents among others. These approaches that target the adult stages of the vectors are summarized below.

#### 3.1.1. Long-Lasting Insecticide-Treated Nets

Long-lasting insecticide-treated nets (LLINs) are factory-made using a fabric treated with an insecticide, usually pyrethroids. Based on unique new fabric technologies and drawbacks associated with the conventional insecticide-treated nets (ITNs), LLINs were invented to withstand repeated washings of up to twenty times under use in field conditions [56]. These LLINs provide physical barriers against host-seeking mosquitoes, and in addition repel or kill the mosquitoes after coming in contact with the chemicals coated on the net fabric [57]. In this way, LLINs offer protection from and control of mosquito-borne diseases both at individual and community levels for a longer duration. Long-lasting insecticide-treated nets are regarded as one of the most effective mosquito control interventions, particularly for preventing malaria [55]. The fact that LLINs do not require retreatment results in lower use of insecticides and consequently a lower emission of the latter into the environment [55,56,58]. The choice of molecules for LLINs relies on whether or not they are effective against the target organism, and proper application may increase their efficacy [57].

In sub-Saharan Africa, where more than 427 million insecticide-treated nets were distributed between 2012 and 2014, morbidity and mortality due to malaria have significantly decreased (by almost 50%) [58]. Long-lasting insecticide-treated nets have also helped to reduce malaria during the past 15 years in pregnant women and children globally [58,59,60,61,62,63]. In a study that was conducted in the southeast of Iran by Soleimani-Ahmadi et al. [63], the prevalence of malaria in groups of LLIN users was dramatically reduced (up to 97%) when compared to groups of LLIN non-users. However, the success of LLINs in averting mosquito-borne diseases depends on several climatic parameters [64]. Access to LLINs also plays a significant role in determining their success [65,66]. Moreover, the effectiveness of LLINs is being hampered by the development of insecticide resistance to the pyrethroids used for their treatment [61]. Hence, there is a need to revamp this vector control tool if mosquito-borne diseases are to be scaled down.

#### 3.1.2. Indoor Residual Spraying

Indoor residual spraying (IRS) involves spraying insecticides inside houses on surfaces that serve as resting places for mosquitoes [67]. *Ae. aegypti*, which primarily rests indoors and feeds on humans, is usually the most affected endophilic species as it is more likely to be reached by IRS than by space sprays [67]. Indoor residual spraying has some drawbacks and flaws, such as the requirement for specialized training, which takes time. Furthermore, it does not stop people from being bitten by mosquitoes [68].

Indoor residual spraying used alone or in combination with larval control can significantly reduce the population of mosquitoes and the disease prevalence [69,70]. The United States President’s Malaria Initiative (PMI) supported the World Health Organization’s 2006 affirmation of IRS’s significance for reducing malaria transmission in 2012 in the Mediterranean region [71]. Consequently, malaria appears to have dwindled as a result of IRS-based malaria eradication programs [72]. In light of insecticide resistance development to pyrethroids among mosquitoes, alternative formulations, such as bendiocarb, are currently being used in the IRS to prevent vectors from developing insecticide resistance [73,74].

#### 3.1.3. Peridomestic Space Spraying

Peridomestic space spraying is the pulverization of insecticides (e.g., dichlorodiphenyltrichloroethane (DDT)) around the home. Through dispersing tiny droplets of insecticides into the air, this method targets only adult mosquitoes and has no direct effect on immature stages (egg, larvae, or pupae) [72]. Peridomestic space spraying is mostly employed in emergency situations to reduce the enormous adult mosquito population [74]. One of the most commonly used approaches in peridomestic spraying is the ultralow volume spray which uses a machine that is hand-held or mounted on a vehicle [74]. Depending on the proportion of the active component in the formulation, the insecticide concentration usually ranges from 2% (pyrethroids) to 95% (organophosphates). The concentration of these active ingredients depends on their toxicity to the target species [72]. Pyrethrin aerial spraying has a substantial influence on the non-target organisms but not on the water quality [75]. Since Dengue and Chikungunya viruses can spread to a small extent by trans-ovarian transmission, mosquitoes that emerge after treatment may still be vectors. Therefore, it is important for the breeding ground of Dengue and Chikungunya vectors to receive subsequent treatments, which should be administered at intervals [72].

#### 3.1.4. Mosquito Repellents

Mosquitoes are primarily drawn to people by the lactic acid and carbon dioxide in their sweat, which their antennae can detect [76]. Mosquito repellents are substances that do not kill mosquitoes but deter them from biting people [76]. The most commonly used repellents obliterate the human aroma, making them efficient insect repellents [72].

Among the synthetic insect repellents, DEET (N, N-diethyl-m-toluamide or N,N-diethyl-3-methylbenzamide)- and IR3535 (3-(N-Butyl-N-acetyl)-aminopropionic acid)-treated clothing works well as a long-lasting insect-repellent [77]. However, the use of synthetic repellents has generated a lot of criticism since it makes mosquitoes resistant to insecticides, harms organisms that are not the intended targets, and poses a threat to the ecosystem [77]. Hence, the use of mosquito repellents from natural sources such as plants, fungi, or bacteria is preferable to the use of chemical repellents for the effective control of mosquitoes and to assure human and environmental safety where endemic mosquito resistance and environmental considerations limit their use [76]. Moreover, nanoparticles have been used as a mosquito repellent when impregnated into cotton fabrics and have shown high efficacy against mosquitoes. This gives them the potential to be used as environmentally friendly mosquito control methods [78].

### 3.2. Biological Control

Each year, promising new environmentally friendly methods are developed to gradually replace the most dangerous and hazardous methods that are used for mosquito control. Some of the encouraging outcomes include the application of biological control programs, such as genetic alteration, biological agents, predatory fish, bacteria, protozoa, nematodes, and fungi, as elaborated below.

#### 3.2.1. Genetic Modification

Based on mass rearing, radiation-mediated sterility, and the release of many male insects into a defined target region to compete for mates with wild males, the sterile insect technique (SIT) is a species-specific and environmentally safe strategy for controlling insect populations. The population tends to drop when wild females mate with the released sterile males [79,80]. However, in some instances, problems associated with mosquito egg production and mass rearing have resulted in the failure of this mosquito control method [81].

#### 3.2.2. Fungi

In recent years, interest in mosquito-killing fungi has revived, mainly due to continuous and increasing levels of insecticide resistance and increasing global risk of mosquito-borne diseases [82]. Indeed, when applied on mosquito resting surfaces, many fungi can infect and kill mosquitoes at the larval and/or adult stages [83]. Among them, one can cite those from the genera *Lagenidium*, *Coelomomyces*, *Entomophthora*, *Culicinomyces*, *Beauveria*, and *Metarhizium*. Notably, *Beauveria bassiana* is one of the most effective against adult *Ae. albopictus* and *Cx. pipiens* mosquitoes [84]. Unfortunately, none of these aforementioned fungi have been specifically developed as larvicidal agents against significant vector species [85,86].

#### 3.2.3. Fish

Fish as predators to control mosquito aquatic stages can be introduced into all probable mosquito breeding sites [87]. Although some failures have been documented in the literature, the employment of native larvivorous fishes is recommended [88]. A small number of species are utilized, principally *Gambusia affinnis* and *Poecilia reticulata* [87]. Other water predators, especially during rainy seasons, may contribute to the decrease in mosquito populations [89,90]. It has also been investigated if naturally occurring, non-biting *Toxorhynchites* species, which display predatory behavior throughout their larval stages, could be used as biological pesticide substitutes (the fourth instar larva is the most predaceous) [91]. Their development for use as biological agents has made significant strides, and they were remarkably effective against a variety of mosquito species, including *Ae. aegypti*, *Ae. albopictus*, and *Cx. quinquefasciatus* [91,92].

#### 3.2.4. Protozoans

Protozoans are single-celled organisms that are found in almost any habitat [93]. The majority of species are free-living, but all higher animals are infected with one or more protozoa species [93]. Among them, one can cite *Chilodonella uncinata* that is a pathogenic parasite with a number of advantageous traits [94]. Indeed, in mosquito larvae, this protozoan results in low to extremely high (25–100%) mortality. When cultivated in vitro, it has a strong reproductive potential, high pathogenicity, and resistance to desiccation. Trans-ovarian transmission allows *Chilodonella uncinata* to spread in the environment through its mosquito host [72].

#### 3.2.5. Bacterial Agents

There are several bacteria and metabolites from bacterial isolates with mosquitocidal properties [95]. However, the use of bacteria as sources of agents for microbial control of mosquitoes received little attention prior to the discovery of *Bacillus thuringiensis* subsp. israelensis (Bti) and *Bacillus sphaericus* (Bs) as effective mosquitocidal agents [96]. Indeed, studies have shown that Bti and Bs alone or in combination are highly efficient and safe for controlling mosquitoes, and they are thought to be safe for non-target organisms co-existing with mosquito larvae [97].

*Bacillus thuringiensis* produces three types of larvicidal proteins during their vegetative phase: Cry (exert intoxication via toxin activation, receptor binding, and pore formation in a suitable larval gut environment), Cyt (cytolytic toxicity) when sporulating (parasporal crystals), and Vip proteins [72]. Some of these proteins, which are toxic to a variety of insect orders and nematodes, are known to cause cancer in humans if mishandled [97]. Compared to Cry toxins, Cyt toxins are less hazardous to mosquito larvae [72]. *Bacillus thuringiensis* species, such as Bti, *B. thuringiensis* var. krustaki, *B. thuringiensis* var. jegathesan (Btjeg), *B. thuringiensis* var. kenyae, and *B. thuringiensis* var. entomocidus cause very high mortality to larval instars of all mosquitoes [98]. Other *Bacillus*-like organisms including *Clostridium bifermentans* (serovar malaysia), *B. circulans*, and *B. laterosporus* exhibit the highest toxicity against dipterans [72].

#### 3.2.6. Insect Growth Regulators

Insect growth regulators (IGRs) are chemicals that mimic or compete with hormones, preventing the development of insects [99]. Among them, methoprene and pyriproxyfen, two juvenile hormone agonists, are becoming more and more popular. Other mosquito growth regulators include novaluron and diflubenzuron [99]. Insect growth regulators are a useful method for reducing mosquito populations because they are selective and have minimal environmental toxicity [72]. Insect growth regulators work well against mosquito larvae and may prevent adult mosquitoes from emerging [100,101]. A recent study has shown that mosquitoes and other pests have become resistant to routinely used IGRs such as methoprene and pyriproxyfen which highlights the need to create new control tools [102].

#### 3.2.7. *Wolbachia* spp.

*Wolbachia* spp. are unicellular Gram-negative bacteria that are present in up to 40% of insects and other arthropods. These bacteria invade the testes or ovaries of the host and alter their reproductive potential [103]. Mosquito symbiont-associated bacteria may be harmful to their host, impairing reproduction and decreasing the vector’s competence [104,105]. Additionally, *Wolbachia* spp. phenotypically cause male killing, cytoplasmic incompatibility, and pathogenicity in mosquitoes [106]. For instance, when a *Wolbachia*-infected male mates with an uninfected female, cytoplasmic incompatibility causes inviable offspring to be produced. This approach was utilized to successfully eradicate *Cx. quinquefasciatus* in Myanmar in the 1960s, and it is now used to target the *Ae. albopictus* strain with triple *Wolbachia* infections [72,107]. The development of *Wolbachia*-superinfected lines with stable infection could increase their role in lowering vector competence in *Ae. aegypti* by blocking the replication of the Dengue virus and preventing the potential emergence of resistance to the bacteria [3,108].

According to several studies, *Wolbachia* spp. prevent the spread of Zika, Dengue, Chikungunya, and yellow fever viruses as well as of *Plasmodium* parasites in *An. stephensi* and *An. gambiae* [109]. A *Wolbachia*-based vector control approach is an urgently needed to complement the biological control programs that are already fairly established, such as the use of Bti.

#### 3.2.8. *Asaia* spp.

*Asaia* is a genus of Gram-negative, aerobic, rod-shaped bacteria from the Acetobacteraceae family that lives in tropical plants. These bacteria can colonize the midgut and male reproductive system of the mosquitoes and spread internally via the hemolymph [110,111,112]. Research has shown that *Asaia* spp. may stimulate the mosquito’s immune system, preventing the growth of malaria parasites [112,113]. The genetic modification of *Asaia* spp. could enable them to colonize new hosts and propagate across wild populations [112,113].

On the other hand, the interruption of the malaria parasite transmission cycle within the vector to stop parasite development before the mosquito becomes infectious is a good way to render vectors ineffective [114]. Paratransgenesis is the easiest method for doing this, which entails creating bacterial strains that can dwell in the midguts of diverse mosquito species and spread quickly across wild mosquito populations [115]. They can then be disseminated in the mosquito population through co-feeding, sexual mating as well as paternal, maternal, and horizontal transmission [111]. A good example is the engineered transgenic *Asaia* spp. that can produce an antiplasmodial effect that makes the mosquito resistant to *Plasmodium berghei*, making it a good candidate for the control of mosquitoes and mosquito-borne diseases [116]. This approach has been used in some *Anopheles* species with promising results in terms of reduced pathogen transmission [117].

#### 3.2.9. Spinosyns

Spinosyns are compounds produced from the fermentation of two species of bacteria genus *Saccharopolyspora*, family Pseudonocardiaceae. *Saccharopolyspora spinosa*, a naturally occurring soil bacterium, is fermented to produce the biopesticide spinosad that contains the insecticides A (C_41_H_65_NO_10_) and D (C_42_H_67_NO_10_) [118]. Spinosad is classified as a group 5 pesticide by the Insecticide Resistance Action Committee (IRAC) and belongs to a novel class of polyketide-macrolide insecticides that act as nicotinic acetylcholine receptor (nAChR) allosteric modulators [72,118,119].

In an early stage of insecticide screening, it was discovered that spinosad was active against a variety of pests in the Lepidoptera, Diptera, Thysanoptera, Coleoptera, Orthoptera, and Hymenoptera orders [72,118]. Among the mosquitoes, the spinosad pesticide has been shown to be effective in reducing larval development in *Ae. aegypti*, *Ae. albopictus*, *An. gambiae*, *An. pseudopunctipennis*, *An. albimanus*, *Cx. Pipiens*, and *Cx. quinquefasicatus* [118,120].

#### 3.2.10. Bacterial-Based Feeding Deterrents and Repellents

Bacterial-based feeding deterrents and repellents are chemical substances that are derived from bacteria not to kill mosquitoes but to deter them from biting people [76]. A combination of chemicals derived from *Xenorhabdus budapestensis* (entomopathogenic-associated bacterium) is shown to deter mosquito species thought to be the most significant disease vectors influencing public health [121]. It has also been shown to be highly effective against mosquitoes, comparable to or better than DEET (N, N-diethyl-m-toluamide) or picaridin [76]. These bacteria deterrents and repellents can be coated on clothes or be made into creams and sprays [72].

### 3.3. Mechanical Control

Mechanical control methods involve the use of traps, sometimes with chemical attractants that are normally given off by mosquito hosts [122,123]. Such chemical attractants usually include carbon dioxide, ammonia, lactic acid, or octenol to attract adult female mosquitoes. Eave tubes and attractive sugar baits are some of the common examples of mechanical control methods [122].

#### 3.3.1. Eave Tubes

The eave tube method comes from the mosquitoes’ native behavioral ecology in sub-Saharan Africa, where they spread malaria [123]. These mosquitoes enter houses through the spaces between the roof and the walls (the eaves) [124]. As a result, closing off the eaves of houses (along with additional window screening) provides a physical barrier that protects residents from malaria [125]. Mosquitoes that enter an eave tube encounter the insecticide-treated netting inside. Eave tubes effectively convert a house into a “lure and kill” device, killing mosquitoes while also acting as a physical barrier to house access [123]. When coverage is high enough, this effect may reduce mosquito populations or change population age structures, both of which would benefit communities [123].

#### 3.3.2. Attractive Sugar Baits

Attractive sugar baits are used to kill both female and male mosquitoes by taking advantage of their sugar-feeding behavior [126]. The mosquitoes are attracted to sugar baits treated with an insecticidal ingredient [126]. In laboratory and field tests, attractive sugar baits were found to be effective against Culicine mosquitoes (*Aedes* species) and sand flies [127]. When used for indoor or outdoor mosquito control, insecticide-treated sugar baits can lower mosquito populations by directly killing mosquitoes that feed on them [128].

Baits are one of the most common solutions, and their combination with other techniques, such as genetic ones, will maximize their efficiency. This is because developing mosquito-specific attractants prevents their effects on non-target species.

### 3.4. Insecticide Resistance among Mosquitoes

With the increased agricultural practices even in urban areas, the amounts of pesticides being applied to the environment have greatly increased. This has possibly favored the development of multiple resistance mechanisms among insect pests and vectors [129].

Due to selective pressure from insecticides, various mosquito species have also alarmingly developed resistance to the various mosquitocides that are currently available for their control [129,130]. This has made mosquito-transmitted infections difficult to control, leading to a great public health and economic burden, especially in tropical African and Asian countries [131,132]. The insecticide resistance of mosquitoes is mediated by various mechanisms [133]. These include mutations in the insecticide’s target site or active metabolites, enzymatic modification of insecticides to produce non-toxic metabolites, and behavioral changes or cuticle thickening. Pyrethroids and DDT have the same target site. The para voltage-gated sodium channel and knockdown resistance (kdr) mutations in this channel can result in DDT and pyrethroid cross-resistance [134].

The insensitive acetylcholinesterase (ace-1) mediates organophosphate and carbamate resistance, resulting in a single nucleotide mutation [133,134]. Metabolic detoxification is typically mediated by gene duplication or transcriptional upregulation of endogenous detoxification enzymes. These include carboxylesterase amplification, primarily through gene duplication, which results in resistance to organophosphates and carbamates. Resistance to pyrethroids and DDT is caused by increased transcription of cytochrome P450-dependent monooxygenases. Additionally, resistance to organophosphates, DDT, and pyrethroids can be caused by the upregulation of glutathione S-transferases (GSTs), which is usually caused by increased transcription rates. There have been numerous reports of insecticide-resistant mosquito species in Africa and Asia, which has resulted in an increase in mosquito-borne infections [133,135,136,137].

## 4. Green Synthesized Plant-Based Metallic Nanoparticles as a Mosquito Control Strategy

As highlighted above, the application of synthetic insecticides (e.g., organochlorines, organophosphates, pyrethroids, and carbamates) has not been very successful due to economic and pharmacological concerns as well as environmental sustainability issues [138]. These concerns and issues include their high costs, increasing resistance in vector species globally, the biological amplification of toxic by-products through the food chain, their harmful effects on humans and other non-target organisms, as well as their non-biodegradable nature [138].

Having this in mind, the development and production of safer, eco-friendly, cost-effective, biodegradable, more efficient, and target-specific plant-based insecticides as a simple and sustainable method of mosquito control has become one of the most effective alternative approaches [139].

According to the literature, more than 2000 plant species that belong to several plant families have been reported for exhibiting ovicidal, larvicidal, pupicidal, adulticidal, and/or repellent activities against different species of mosquitoes [138,139]. Notably, the insecticidal effects of these plants vary not only according to (i) their species, (ii) their geographical origin and seasonal variety, (iii) the parts used (fruits, leaves, stems, barks, roots, etc.), (iv) their age (senescent, mature or young), (v) the procedure of extraction and the polarity of the used solvents, and (vi) the photosensitivity of some of their compounds, but also (vii) the mosquito vector species and (viii) their developmental stages [138]. The majority of plant extracts with mosquitocidal activities comparable to those of chemical pesticides currently marketed (i.e., with LC_50_ values ˂ 30 ppm) were mainly prepared using moderately polar and nonpolar solvents [138,140]. This is the case of (i) petroleum ether extracts of *Artemisia annua*, *Argemone mexicana*, *Aloe barbadensi*, *Jatropha curcas*, *Piper nigrum*, *Euphorbia tirucalli*, *Ocimum basilicum*, and *Piper longum* [140,141,142,143], (ii) hexane extracts of *Momordica charantia*, *Khaya senegalensis*, *Cybistax antisyphilitica*, *Eucalyptus citriodora*, and *Solanum nigrum* [144,145,146], (iii) carbon tetrachloride extract of *Aloe barbadensis* [147], (iv) chloroform extracts of *Plumbago zeylanica*, *Plumbago dawei*, *Plumbago stenophylla*, *Nyctanthes arbortristis*, and *Cassia obtusifolia* [138,148], (v) methanol extracts of *Atlantia monophylla*, *Ocimum gratissimum*, *Solenostemma argel*, *Centella asiatica*, *Cassia tora*, *Aristolochia saccata*, *Annona squamosa*, and *Chamaecyparis obtuse* [149,150], (vi) mixture of chloroform and methanol extracts of *Solanum villosum* and *Cestrum diurnum* [151,152,153,154], (vii) ethanol extracts of *Azadirachta indica*, *Ocimum gratissimum*, *Piper longum*, *Piper ribesoides*, *Piper sarmentosum*, *Annona crassiflora*, *Annona glabra*, *Denis* spp., *Erythrina mulungu*, and *Eclipta paniculata* [138,155,156,157,158], (viii) benzene extracts of *Citrullus vulgaris* and *Acalypha indica* [157,158,159,160], (ix) ethyl acetate extracts of *Aloe turkanensis*, *Solanum nigrum*, and *Annona squamosa* [159,160,161,162], and (x) acetone extract of *Ageratum conyzoides* [163].

By contrast, it was found that the extraction with water as solvent represents less than almost 40% of studies related to the mosquitocidal evaluation of botanical-based insecticides [138,140,164]. However, only half of these aqueous extracts have a 50% lethal concentration (LC_50_) lower than 30 ppm while they are considered eco-friendly and scarcely toxic to humans (and other vertebrates). In addition, they can be readily prepared and employed by poor populations worldwide, without extremely high costs. One can cite (i) ethanol–water extracts of *Artemisia annua*, *Cassia fistuala*, and *Centella asiatica* [140], (ii) aqueous extracts of *Carica papaya*, *Murraya paniculata*, and *Cleistanthus collinus* [138,162] as well as (iii) steam distilled extracts of *Paullinia claviger, Tradescintia zebrine*, *Lantana camara*, *Artemisia vulgaris*, *Platycladus orientalis*, *Argemone mexicana*, *Allium sativum*, *Pimpinella anisum*, *Lippia berlandieri*, *Clinopodium macrostemum*, *Piper betle*, *Artemisia annua*, *Cassia fistula*, *Centella asiatica*, *Eucalytus globulus*, and *Piper retrofractum* [165,166,167,168,169,170,171,172,173,174].

Of note, several essential oils from plants also exhibited pronounced insecticidal and repellent activities against mosquitoes. However, the fact that their dispersion in a water environment requires the use of synthetic surfactants constitutes a major weakness [175,176].

The primary benefit of the aforementioned plants and their homemade preparations in the fight against malaria and other mosquito-borne diseases is the physical prevention of mosquito ingress into the house [176]. However, some promising plants with highly effective toxic properties against mosquito vectors have failed to provide marketable products due to factors such as the insufficient stability of some of their bioactive phytocompounds in the environment, the low persistence of their insecticide effects, and the lengthy and expensive registration process for botanical insecticide formulations [176].

Given the foregoing, it is critical to seek appropriate formulations and/or stabilization methods for extending the effectiveness and persistence of plant-based insecticides and products derived from them [177]. One of the most promising approaches for this purpose is the use of plant extracts and their selected constituents as reducing and capping/stabilizing agents in the synthesis of green metal and metal oxide nanoparticles. Notably, this approach refers to “green synthesis of metallic nanoparticles”, i.e., the preparation of different metallic nanoparticles from bioactive compounds such as plants and plant extracts, microorganisms, and enzymes [178]. Green synthesis is a biological method that respects the environment by avoiding the handling of toxic chemical compounds. It requires the use of a “green” solvent, an ecological reducing agent, and eventually a non-toxic stabilizing agent [179]. Additionally, green nanosynthesis enables the use of low-cost agricultural wastes (e.g., fruit peels and extraction process residues), invertebrate byproducts (e.g., chitosan from crab shells), and fungal extracellular filtrates as reliable sources of reducing agents [180].

Briefly, to produce green synthesized plant-based metallic nanoparticles, different concentrations of (fresh) plant extracts (and/or other natural sources of reducing agents) are mixed with different concentrations of metal precursor solutions (silver nitrate, cupric nitrate, gold chloride, palladium chloride, copper sulfate, etc.) under different volume ratios and reaction conditions (temperature, pH, stirring speed, etc.) [181,182]. Indeed, these parameters can affect the rate and stability of nanoparticle formation. Due to the presence of their functional groups (carboxylic acids, ketones, amides, etc.), alkaloids, polyphenols, terpenoids, sugars, and many other biologically active phytocompounds are able to reduce metal ions very quickly [183]. The obtained metallic nanoparticles are then purified by centrifugation and the recovered pellet is dried in the oven [184].

In fact, one of the most important requirements to fulfill in the synthesis of these nanoparticles as a vector control strategy is their absence of toxicity based on the employment of plants with longstanding uses in traditional medicine [176,185,186,187]. Moreover, the botanical blends of phytochemicals (e.g., phenolics, alkaloids, essential oils, etc.) that behave as capping agents may act concertedly with each other and with the reduced metals, thereby ensuring there is very little chance of mosquito vectors developing resistance to them [176,187].

It is worth noting that green nano(bio)technology produces nanoparticles with sizes typically smaller than 100 nm and a high surface-to-volume ratio, which accounts for their exceptional pharmacological performance [188]. They can also release a slow but constant number of ions over time, which is especially useful in liquid and sol-gel environments [176,189]. In comparison to the traditional chemical and physical synthesis routes currently used in nanotechnology, green synthesis of metal and metal oxide nanoparticles is typically inexpensive, quick, and does not necessitate high pressure, energy, temperature, or the use of highly toxic chemicals [188,190].

In this section, the potential toxicity of various green synthesized plant-based metallic nanoparticles against mosquito vectors is summarized. We have focused on the repellent, ovicidal, larvicidal, pupicidal, and adulticidal activities of these nanoparticles.

### 4.1. Green Metallic Nanoparticles as Repellents

The repellent activity of green metallic nanoparticles (MNPs) is assessed using the cage arm test, which is probably the first well-planned laboratory test of mosquito repellency [191]. Briefly, for the repellent evaluation, a fixed number of non-blood-fed female mosquitoes (i.e., that are starved during the previous 12 h) are placed in a test cage at 25–29 °C with a relative humidity of 60–90% [191]. A volume of the candidate repellent contained in a diluent solvent is then applied to one volunteer’s forearm (and the standard solution on another forearm). The time elapsed between the application of the repellents and the first mosquito bite in the treated arm was recorded and considered as protection time [192,193,194]. The percentage of mosquito bites is calculated for each trial using the following formula:$$\%\;\mathrm{biting}=\frac{\mathrm{NBUA}-\mathrm{NBTA}}{\mathrm{NBUA}}\times 100$$
where NBUA = number of bites received by the untreated arm, and NBTA = number of bites received by the treated arm [192].

Moreover, to calculate the repellency percentage, the number of mosquitoes that land or suck blood on the skin is recorded in a determined time. The formula for repellency percentage is as follows:$$\mathrm{Percentage}\;\mathrm{repellent}=\frac{C-T\;}{C}\times 100\;$$
where C represents the number of unrepelled mosquitoes by the negative control and T represents the number of unrepelled mosquitoes by the green synthesized MNPs [193,195].

This test is performed with at least five different concentrations of MNPs. Notably, different adaptations of the cage arm test based on the variations in mosquito cage size (40 × 40 × 40 cm^3^, 40 × 40 × 30 cm^3^, 50 × 50 × 50 cm^3^, 80 × 40 × 40 cm^3^, etc.), number of mosquitoes (10–250) and their age (3–5 days) are reported in the literature [194,196,197]. Considering these different variables, the World Health Organization (WHO) has given some guidelines to standardize the conditions of the repellent assay [198].

To the best of our knowledge, only two records concerning the repellent potential of MNPs are available. The first one is related to *Morinda citrifiolia*-fabricated silver nanoparticles (100 mg/mL) that exhibited 53–60% repellency against *An. maculatus*, *An. Gambiae*, and *Cx. quinquefasciatus*, and 35% repellency against *Ae. aegypti*. Used as control, the extract of *Morinda citrifolia* (15%, *w*/*v*) showed repellent activities two to nine times smaller than those of nanoparticles [193]. The second one concerns *Artemisia vulgaris*-synthesized gold nanoparticles that exhibited a protection time of 60% against *Ae. aegypti* [192].

### 4.2. Green Metallic Nanoparticles as Ovicides

Ovicidal tests for green metallic nanoparticles are based on different non-standardized procedures [199,200,201]. Briefly, numerous mosquito egg rafts of different ages (0, 4, 8, 12, and 24 h) are laid in cups containing the tested nanoparticles (or distilled water as control) for 36 h. A predetermined number of eggs are then transferred into new cups containing distilled water where they are allowed to hatch for a total period of 120 h after oviposition. The test is conducted in a holding room at 28 ± 1 °C, 35–45% relative humidity, and 14:10 h light:dark photoperiods with 1 h dawn and dusk periods [199,202,203]. The ovicidal activity of MNPs is determined by calculating the hatching rate, i.e., the percentage of the hatched larvae by the eggs initially tested.

It should be noted that the age of mosquitoes is an important factor to standardize for allowing a rational comparison between different experiments [199,203]. Indeed, eggs over 4 h old were no longer susceptible to certain plant-based products [199]. Moreover, the number of eggs used for a certain volume of tested solution should also be standardized to better compare the different ovicidal assays [200].

The literature includes several studies on the ovicidal toxicity of green synthesized nanoparticles. Some details related to the ovicidal properties of MNPs are given in Table 1 and Appendix A. In general, mosquito eggs showed higher resistance to nanoparticle-based treatments than larvae.

### 4.3. Green Metallic Nanoparticles with Larvicidal and Pupicidal Properties

Guidelines for larvicidal and pupicidal tests developed by the WHO include the rearing of larvae and the evaluation of their potential [199,200,231]. Briefly, larvae rearing includes the collection and hatching of eggs in cups containing dechlorinated, tap, or distilled water and larval food at 25 ± 1 °C. After the eggs hatch, the first instar larvae are regularly fed at 1- or 2-day intervals and monitored to assess their development. Once larvae reach the desired instar, the tests can be conducted. Hatching time and the time required to reach the desired instar depends on the species. For example, *Ae. aegypti* takes only about 12 h for hatching whereas *Culex* spp. can take 1–2 days [198,232]. Notably, many researchers collect a batch of larvae directly from nature and select the desired instar for the larvicidal test [226,233,234].

For testing the larvicidal and pupicidal activity of green metal and metal oxide nanoparticles, the WHO procedures are generally used [230,235,236]. In these tests, batches of 20 to 25 late 3rd instar or young 4th instar larvae are immersed into different cups or beakers of 500 mL containing 250 mL of tested solutions prepared with dechlorinated, distilled, or tap water at 25 ± 2 °C. At least five concentrations of MNPs must be tested with four different cup tests for each concentration. The whole test is replicated at least three times with larvae coming from different rearing batches. The mortality percentage is recorded 24 h later (and 48 h if necessary). Larvae (and pupae) are considered dead if they cannot be induced to move when probed with a needle in the siphon or cervical region [140,198,232].

A curve of mortality percentage as a function of the logarithms of concentrations tested is plotted, and the LC_50_ is computed. In general and depending on the mosquito species, a larvicide with an LC_50_ lower than 10 µg/mL (10 ppm) after 24 h of exposition is considered highly effective [140,232,237].

It should be noted that modifications to the WHO standardized procedures are common. These variations concern mainly the operating conditions during the assay such as the volume of solutions containing the larvae to be tested (100–250 mL) and their concentration (number of larvae per mL) [175,224,231,238,239,240]. In addition, some studies report the use of first and second instar larvae in the test even though late third and young fourth instar larvae are recommended [241,242,243,244].

Another important factor is the provision of larval food during the experiments, which is sometimes done [235,245,246,247,248,249] and sometimes not [247,250,251]. Of note, fed larvae could be considered more resistant than starved ones, thus raising questions about the equivalence of two assays during a comparison.

Finally, the information on how the death of larvae is determined is sometimes not reported, leaving no way of knowing if this was done following the WHO recommendation or not [226,252,253,254,255]. Notably, the literature has also reported other larvicidal assay procedures for metal nanoparticles completely different from the WHO standardized procedures, but they are not very common [244,256,257]. Table 1 and Appendix A summarize some green sources, and the larvicidal and pupicidal activities of the developed nanoparticles. As shown in Figure 1, based on all the data listed in Table 1 and Appendix A, 82.2% of studies that concern the control of mosquito vectors by MNPs focused on their larvicidal and pupicidal potentials. It also should be noted that 42.1% of the studies concentrated their attention on silver nanoparticles, of which 59.8% were obtained from leaf extracts (see Figure 2). The most tested mosquito vectors were *Ae. aegypti* (29.6%), *Cx. quiquefasciatus* (20.7%), and *An. stephensi* (18.6%). Interestingly, 45.5% of the MNPs exhibited LC_50_ values ranging between 1 and 10 ppm, and 21.8% between 11 and 30 ppm (see Figure 1). In addition, the tested mosquito species come either from laboratory strains (51%) or from wild populations (37%).

### 4.4. Green Metallic Nanoparticles as Adulticides

The adulticidal assay consists of bringing adult mosquitoes into contact with filter papers (12 × 15 cm) previously impregnated with a volume of MNPs in solution and evaluating their mortality [232,238,258]. Briefly, batches of 20 to 25 mosquitoes 3 to 5 days old that are non-blood-fed (but fed with sugar or starved for 6 h) are placed in different tubes containing the treated filter papers. After one hour of exposition, mosquitoes are returned to their holding tubes and the mortality rate is recorded 24 h later. The curve of mortality rate as a function of logarithms of concentrations is then plotted. In general, an insecticide is considered effective if the LD_50_ is inferior or equal to 0.16 mg/cm^2^ [232,238,258].

To perform this test, researchers can use the WHO kit or homemade equipment that has similar characteristics to that of the WHO [121]. This test is carried out with serial concentrations of MNPs and replicated at least three times using mosquitoes from different rearing batches to allow for inter-batch variability. A summary of the adulticidal properties of green synthesized metallic nanoparticles is shown in Table 1 and Appendix A.

### 4.5. Nanoparticles as Mosquito Longevity and Fecundity Reducers and Mosquito Oviposition Deterrents

The influence of green nanoparticle-based treatments on adult longevity and female fecundity of mosquito vectors was recently highlighted. Indeed, the treatment of *Ae. aegypti* with silver nanoparticles from *Hypnea musciformis* (50 ppm) exhibited a reduction of 1.5 to 3 times in male and female longevity, and female fecundity [259]. Similar results were obtained with *An. stephensi* treated with *Pteridium aquilinum*-synthesized silver nanoparticles (50 ppm) [260].

Additionally, it was reported that 10–12 ppm of silver nanoparticles fabricated using the extract of *Sargassum muticum* [203], *Dicranopteris linearis* [261], or *Rubus ellipticus* [262] led to a reduction in mosquito oviposition rates of more than 70% in *Ae. aegypti*, *An. stephensi*, and/or *Cx. quinquefasciatus*.

Overall, limited knowledge is available about the influence of green metallic nanoparticle-based treatments on mosquito longevity and fecundity as well as the oviposition behavior (attraction/deterrence) of mosquito vectors [263]. Further research is urgently needed.

### 4.6. Possible Mechanisms of Action of Mosquitocidal Green Metallic Nanoparticles

As mentioned above, green metallic nanoparticles show toxic effects on larvae and other mosquito stages of growth (pupae, adults, and eggs). However, to date, the exact physiological and molecular mechanisms of these nanoparticles against mosquitoes are still unknown [264,265]. Of note, these mechanisms of action would be complex and multiple, especially since they would be linked to the nanomaterials themselves, to the ions they generate, and to the phytoconstituents of capping agents.

Nevertheless, it is accepted that green MNPs may act as Trojan horses and penetrate mosquitoes through the exoskeleton while bypassing the cell barrier and then generating toxic ions to cause cell damage [266]. Once inside the intracellular space, green fabricated nanoparticles can bind to sulfur of proteins or phosphorus of deoxyribonucleic acids (DNA), and thus cause a rapid denaturation of organelles and proteins [267,268,269]. In addition, a decrease in the activity of acetylcholinesterase and α- and β-carboxyesterases in fourth instar larvae of *Ae. albopictus* and *Cx. pipiens* was observed after exposure of phytonanoparticles prepared using an extract of *Cassia fistula* [268,270]. Green metallic nanoparticles may also work by inducing oxidative stress in mosquitoes. All these mechanisms of toxicity depend on the size, shape, and charge of the greenly synthesized MNPs [267].

Additionally, numerous studies have reported that the toxicity of green fabricated MNPs on mosquitoes may result in noticeable morphological and histological abnormalities or changes [136,177,264,271,272]. For instance, *Ae. aegypti* larvae exposed to green silver or gold nanoparticles exhibited a disruption of midgut epithelial cells and cortex damage [7,271,273,274]. Moreover, shrinkage of the abdominal region, change in the shape of the thorax, damage to the midgut, loss of lateral hair, and anal gills were reported in *Ae. aegypti* third instar larvae after exposure to green encapsulated zinc oxide nanoparticles [274].

Concerning the phytocompounds (flavonoids, coumarins, anthraquinones, alkaloids, etc.) that act as capping (and stabilizing) agents, their possible mechanisms of action against mosquito larvae are summarized in the literature [138,139,275]. They include inhibition of the cholinergic system, binding to gamma-aminobutyric acid (GABA) receptors, interaction with cytochrome P450, modulation of the octopaminergic system, inhibition of nicotinamide adenine dinucleotide hydrogen (NADH) ubiquinone oxidoreductase, reduction in protein level, inhibition of glutathione S-transferase, blocking of sterol carrying proteins, generation of reactive oxygen species (ROS), antifeedant activity, phototoxicity, neuronal toxicity, midgut damage, etc. [138,139,275,276]. Therefore, there is a real need for further investigations to fully uncover the details related to the real mechanism of these nanoparticles against any mosquito developmental instar, especially those related to mosquito eggs and adult mosquitoes.

### 4.7. Non-Target Effects of Mosquitocidal Green Metallic Nanoparticles

The growing pollution of the environment due to the accumulation of conventional pesticides has become a major concern. The availability of metallic nanoparticles of plant origin with larvicidal, pupicidal, ovicidal, and adulticidal properties may therefore be useful in reducing the risk of ecotoxicity [276,277]. Indeed, an important characteristic of these green metallic nanolarvicides is that they have lower residual pollution without noticeable toxicity effects against non-target aquatic organisms. It has been reported that a 48–72 h exposure of an aquatic environment to doses of green metallic nanoparticles toxic to mosquito species has no evident toxicity against *Poecilia reticulata* fish [278,279,280]. Similarly, at lethal concentrations to mosquito larvae, silver metallic nanoparticles synthesized using plant extracts showed no obvious minimal biotoxicity to non-target mosquito predators such as *Anisops bouvieri*, *Diplonychus indicus*, *Gambusia affinis*, *Toxorhynchites splendens*, *Diplonychus annulatum*, and *Mesocyclops pehpeiensis* [279,280,281,282,283,284,285].

Notably, in an aquatic environment contaminated by sub-lethal doses of green fabricated gold and silver nanoparticles, the predation efficiency of mosquito natural enemies (*Mesocyclops aspericornis*, *Gambusia affinis*, and *Carassius auratus*) was boosted compared to their standard predation rates, thereby contributing to the control of *Anopheles*, *Aedes*, and *Culex* larval populations in fish-, copepod-, and tadpole-based control programs [248,286,287].

Hence, these nanoscale forms with properties of effectiveness at lower concentration and targeted specificity with lower residual pollution in the environment improves their effectiveness compared to conventional pesticides [288,289,290]. Nevertheless, due to moderate information and limited evidence of the absence of toxicity of green synthesized mosquitocidal nanoparticles towards aquatic non-target species and their environments [290], other studies, such as acute toxicity and genotoxicity, should be carried out to confirm these observations. The persistence of green metallic nanoparticles in the environment should also be determined in the future.

## 5. Conclusions and Perspectives

Among the several existing vector control tools, none of them is fully successful, thereby resulting in the development of insecticide resistance, the resurgence of mosquito populations, the onset of multiple adverse effects on humans and other non-target organisms, and/or the disruption of natural ecosystems.

It is because of this that the idea of developing novel eco-friendly strategies (plant extracts and plant-based nanotechnologies) to manage mosquito vectors was born. Despite their often high LC_50_ (˃100 ppm), plant extracts are an effective solution for reducing adverse effects on the environment and human health. Therefore, interest in plant-based metallic nanoparticles has grown in recent years, as their green synthesis is cheap, single-step, and does not require high pressure, energy, temperature, or the use of highly toxic chemicals. Highly effective green MNPs also contain a potent source of bioactive phytochemicals that are safe and biodegradable into non-toxic by-products, which could be screened for ovicidal, larvicidal, pupicidal, adulticidal, and repellent activities. Moreover, green fabricated nanoinsecticides may also lack off-target effects on non-target aquatic organisms. Altogether, based on the literature reported in this review, green synthesized plant-based metallic nanoparticles represent a promising strategy in the control of mosquito-borne diseases.

## Figures and Tables

**Figure 1 insects-14-00221-f001:**
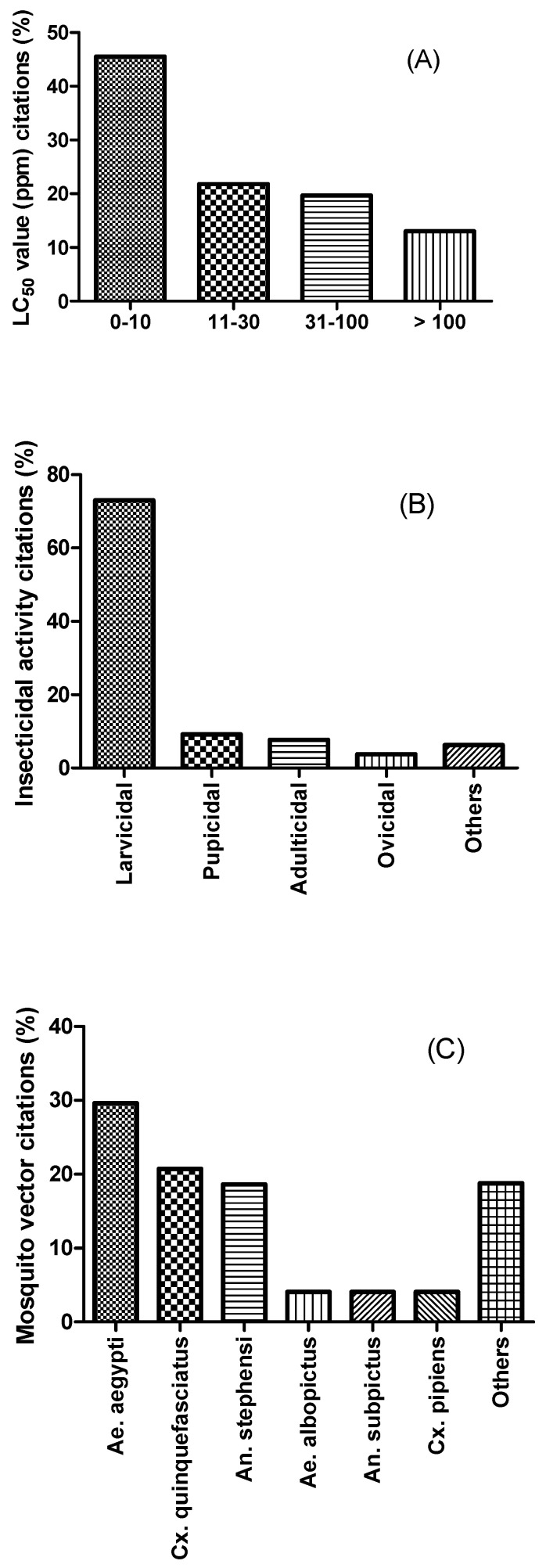
Pooled data from studies related to mosquitocidal MNPs. (**A**) Percentage of LC_50_ value (ppm) citations (*n* = 122). (**B**) Percentage of insecticidal activity citations (*n* = 121). Others include mosquito longevity and fecundity reduction as well as mosquito oviposition deterrent activity. (**C**) Percentage of mosquito vector citations (*n* = 126). Others include *Cx. gelidus*, *Cx. vishnui*, *Ae. vittatus*, etc.

**Figure 2 insects-14-00221-f002:**
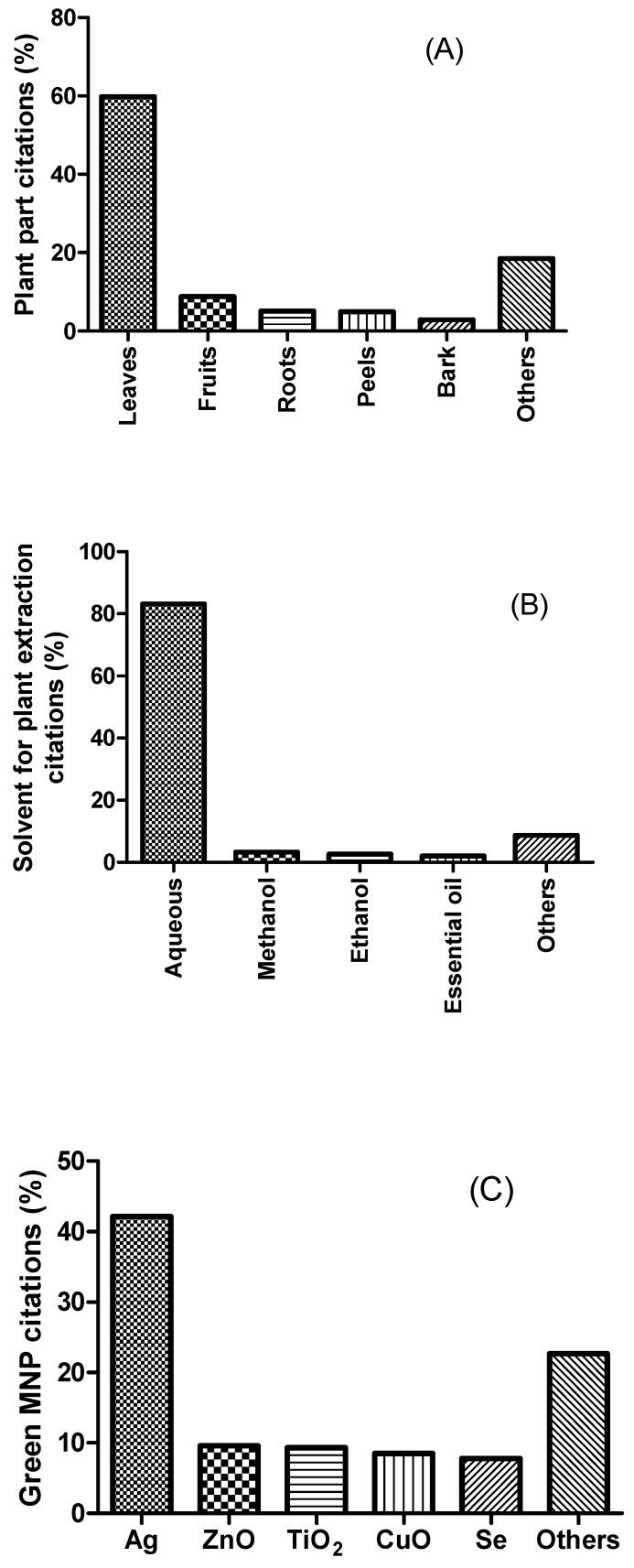
Pooled data from studies related to green MNPs with mosquitocidal properties. (**A**) Percentage of plant part citations (*n* = 124). Others include rhizomes, tubers, petals, seeds, latex, etc. (**B**) Percentage of solvent for plant extraction citations (*n* = 129). Others include ethyl acetate, hexane, petroleum ether, etc. (**C**) Percentage of green MNP citations (*n* = 124). Others include gold, cobalt, nickel, palladium, cadmium, and iron.

**Table 1 insects-14-00221-t001:** Green metallic nanoparticles with mosquitocidal properties.

	Materials		Operative Conditions for Synthesis	Shape and Size	Activity and Species	Origin of Mosquitoes	LC_50_ ^a^	Ref
	Plants	Metal Precursors						
1	*Syzgium cumini* seed (aqueous extract 5%)	Zn(CH_3_COO)_2_ 1 mM	Ratio 20:80 Incubation time: 45 min Temperature: 60 °C	Spherical, 50–60 nm (TEM, SEM) ^b,c^	Larvicidal and Ovicidal (*Ae. aegypti*)	Wild populations	51.94	[201]
2	*Ficus racemosa*, bark (aqueous extract 10%)	AgNO_3_ 1 mM	Ratio 8:2 Incubation time: 10 min Temperature: room	Cylindrical, 250.06 nm (TEM)	Larvicidal (*Cx. quinquefasciatus* and *Cx. gelidus*)	Wild populations	12.00 11.21	[204]
3	*Piper longum* L. leaves (aqueous, chloroformic, ethyl acetate, hexane, methanolic extracts 10%)	AgNO_3_ 1 mM	Ratio 9:1 Incubation time: 20 min Temperature: 40 ± 2 °C pH 7.5	Spherical, 25–32 nm (HR-TEM, FE-SEM) ^d,e^	Larvicidal (*Ae. aegypti, An. stephensi* and *Cx quinquefasciatus*)	Laboratory strains	8.97	[205]
4	*Holarrhena* *antidysenterica (L.*) bark (aqueous extract 4%)	AgNO_3_ 1 mM	Ratio 9:1 Incubation time: 5 min Temperature: 50 ± 2 °C pH 7.5	Spherical, 32 nm (FE-SEM, TEM)	Larvicidal (*Ae. aegypti L.* and *Cx. quinquefasciatus*)	Laboratory strains	9.3	[206]
5	*Ammannia baccifera* aerial parts (aqueous extract 8%)	AgNO_3_ 1 mM	Ratio 40:3 Incubation time: 30 min Temperature: room	Triangular and hexagonal, 10–30 nm (TEM, SEM)	Larvicidal (*An. subpictus* and *Cx. quinquefasciatus*)	Laboratory strains	29.54 22.32	[207]
6	*Tridax procumbens* leaves (aqueous extract 10%)	CuSO_4_ 1 mM	Ratio 4:9 Incubation time: 4 h Temperature: 80 °C	Spherical 16 nm (FE-SEM)	Larvicidal (*Ae. aegypti*)	Wild populations	4.21	[208]
7	*Grewia asiatica* leaves (aqueous extract 10%)	CuSO_4_.5H_2_O 1 mM	Ratio1:9 Incubation time: 60 min Temperature: 70 °C	Spherical 60–80 nm (SEM)	Larvicidal (*Ae. aegypti*)	NM	100	[209]
8	*Annona squamosa* seeds (aqueous extract 15%)	CuSO_4_ 1 mM	Ratio 1:3 Temperature: 28 ± 2 °C Incubation time: 24 h	Spherical 5.99–24.48 nm (TEM)	Larvicidal (*An. stephensi*)	Laboratory strains	170	[210]
9	*Mangifera indica*, leaves (aqueous extract 10%)	TiO(OH)_2_ 5 mM	Ratio 1:17 Incubation time: 24 h Temperature: 37 °C	Spherical 30 ± 5 nm (TEM)	Larvicidal (*An. subpictus* and *Cx. quinquefasciatus*)	Wild populations	5.84 4.31	[211]
10	*Momordica charantia* leaves (aqueous extract 10%)	TiCl_4_ 5 mM	Ratio1:4 Incubation time: 15 min Temperature: room	Irregular 70 nm (HR-TEM)	Larvicidal and pupicidal (*An. stephensi*)	Wild populations	3.43	[212]
11	*Morinda citrifolia* roots (aqueous extract 8%)	TiO(OH)_2_ 5 mM	Ratio 1:4 Incubation time: 4 h Temperature: 50 °C	Spherical, oval, and triangular 20.46–39.20 nm (TEM)	Larvicidal (*An. stephensi*, *Ae. Aegypti* and *Cx. quinquefasciatus*)	NM	5.03 16.29 21.64	[213]
12	*Solanum trilobatum* leaves (aqueous extract 10%)	TiO(OH)_2_ 5 mM	Ratio 1:4 Incubation time: 24 h Temperature: Room	Spherical ~70 nm (SEM)	Larvicidal (*An. subpictus*)	Wild populations	0.97 0.99 0.99	[214]
13	*Vitex negundo*, leaves (aqueous extract 10%)	TiCl_4_ 5 mM	Ratio 1:4 Incubation time: 15 min Temperature: Room	Spherical ~93.3 nm (SEM)	Larvicidal (*An. subpictus* and *Cx. quinquefasciatus*)	Wild populations	7.52 7.23	[215]
14	*Clausena dentate*, leaves (aqueous extract 10%)	H_2_SeO_3_ 1 mM	Ratio 3:22 Incubation time: 24 h Temperature: 37 °C	Spherical, 46–78 nm (SEM)	Larvicidal (*An. stephensi, Ae. aegypti* and *Cx. quinquefasciatus*)	Laboratory strains	240.71 104.13 99.60	[216]
15	*Ceropegia bulbosa*, tuber (aqueous extract 10%)	H_2_SeO_3_ 40 mM	Ratio 1:4 Incubation time: 24 h Temperature: 37 °C	Spherical 55.9 nm (HR-TEM)	Larvicidal (*An. albopictus*)	Laboratory strains	250	[217]
16	*Nigella sativa*, seed (aqueous extract 5%)	H_2_SeO_3_ 0,01 mM	Ratio 1:1 Incubation time: 1 h Temperature: 60 °C	Clusters, 55–75 nm (SEM)	Larvicidal (*Cx. pipiens*)	Laboratory strains	17.39	[218]
17	*Nilgirianthus ciliates* leaves (aqueous extract 5%)	H_2_SeO_3_ 30 mM	Ratio 1:4 Incubation time: 24 h Temperature: 30 ± 2 °C	Spherical 100 nm (FE-SEM)	Larvicidal (*Ae. aegypti*)	Wild populations	0.92	[219]
18	*Opuncia ficus-indica* peel (aqueous extract 10%)	Na_2_SeO_3_ 2 mM	Ratio 1:9 Incubation time: 24 h Temperature: 37 °C	Spherical 10–87 nm (TEM)	Larvicidal (*Cx. pipiens*)	Laboratory strains	75.41	[220]
19	*Ulva lactuta* seaweed (aqueous extract 10%)	Zn(CH_3_COO)_2_. 2H_2_O 1 mM	Ratio 5:95 Incubation time: 4 h Temperature: 70 °C	Sponge-like 10–50 nm (TEM)	Larvicidal (*Ae. aegypti*)	Wild populations	38	[221]
20	*Myristica fragrans* fruit (aqueous extract 10%)	Zn(NO_3_)_2_.6H_2_O	Ratio 6% (*w*/*v*) Incubation time: 60 °C Temperature: 2 h	Semispherical, hexagonal 43.3–83.1 nm (TEM, SEM, DLS) ^f^	Larvicidal (*Ae. aegypti*)	NM	5	[222]
21	*Cocos nucifera* fruits (methanol extract 4%)	Pd(OAc)_2_ 1 mM	Ratio 1:1 Incubation time: 6 h Temperature: 60 °C	Spherical 32 ± 3 nm (TEM)	Larvicidal and ovicidal (*Ae. aegypti*)	Wild populations	288.88	[223]
22	*Citrus limon* leaves (aqueous extract 10%)	PdCl_2_ 1 mM	Ratio 1:9 Incubation time: 24 h Temperature 29 °C	Spherical, 1.5–18.5 nm (TEM)	Larvicidal (*An. stephensi*)	Wild populations	5.12–10.83	[224]
23	*Nephrolepis exaltata* whole plant (aqueous extract 10%)	FeCl_3_.6H_2_O 0.01 M	Ratio 9:1 Incubation time: 2 h Temperature: 60 °C pH: 8	Spherical 30–70 nm (TEM)	Larvicidal (*An. stephensi*)	Wild populations	25	[225]
24	*Aegle marmelos*, leaves (aqueous extract 10%)	NiCl_2_ 1 mM	Ratio 1:4 Incubation time: 5 h Temperature: 60 °C	Triangular 80–100 nm (SEM)	Larvicidal (*An. stephensi*, *Ae. aegypti* and *Cx. quinquefasciatus*)	Wild populations	534.83 595.23 520.83	[226]
25	*Cocos nucifera*, fruits (aqueous extract 1%)	Ni (OAc)_2_ 1 mM	Ratio 1:4 Incubation time: 7 h Temperature: 60 °C	Cubical 47 nm (TEM)	Larvicidal (*Ae. aegypti*)	Wild populations	259.24	[227]
26	*Artemisia vulgaris*, leaves (aqueous extract 10%)	HAuCl_4_ 1 mM	Ratio 1:9 Incubation time: 24 h Temperature: 37 °C	Spherical, triangular, and hexagonal 50–100 nm (TEM)	Larvicidal (*Ae. aegypti*)	Laboratory strains	74.42	[228]
27	*Moringa oleifera*, leaves (aqueous extract 2.5%)	HAuCl_4_.3H_2_O 1 mM	Ratio 1:10 Incubation time: 24 h Temperature: 37 °C	Spherical, oval, triangular, and pentagonal 10–60 nm (TEM)	Larvicidal (*Cx. quinquefasciatus*)	Laboratory strains	8.24	[229]
28	*Cymbopogon citrates*, leaves (aqueous extract 5%)	HAuCl_4_ 1 mM	Ratio 1:10 Incubation time: 72 h Temperature: 25 °C	Spherical, triangular, hexagonal, and rod 20–50 nm (TEM)	Larvicidal (*An. stephensi* and *Ae. aegypti*)	Wild populations	18.50–41.52	[230]
29	*Curcuma zedoaria*, roots (essential oil)	AgNO_3_ 5 mM	Ratio NM Incubation time: NM Temperature: NM pH 7 (NaOH 0.1%)	Globular 92.44 nm (SEM)	Larvicidal (*Cx. quinquefasciatus*)	Laboratory strains and wild populations	36.32	[231]
30	*Halodule uninervis* Leaves (aqueous extract 10%)	AgNO_3_ 1 mM	NM ^g^	Cubic, 20–35 nm (SEM)	Larvicidal (*Ae. aegypti*)	Wild populations	12.56	[136]

^a^ LC_50_: 50% lethal concentration (in ppm unless otherwise stated); ^b^ TEM: transmission electron microscopy; ^c^ SEM: scanning electron microscopy; ^d^ HR-TEM: high-resolution transmission electron microscopy; ^e^ FE-SEM: field emission scanning electron microscopy; ^f^ DLS: dynamic light scattering; ^g^ NM: not mentioned.

## Data Availability

All data generated during this study are included in this article.

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
