# Peer review of "Mosquito-Borne Diseases and Their Control Strategies: An Overview Focused on Green Synthesized Plant-Based Metallic Nanoparticles"

_insects, 2023, doi:10.3390/insects14030221_

Round 1

Reviewer 1 Report

insects-2095625

Title: Mosquito-borne diseases and their control strategies: an overview focused on green-synthesized plant-based metallic nanoparticles

Comments to the authors

The paper from Onen et al. makes a review on green-synthesized plant-based metallic nanoparticles. The application of nanotechnological approach towards the control of the harmful vectors remains an important strategy towards the vector control. Thus, the advancement in the nanotechnology provides several methods to counter increasing concern of the vector-borne diseases. In this context, the application of nanotechnology to formulate nanopesticides with the greener approach is recommended.

Green synthesis of metal and metal oxide nanoparticles is known to be low-cost, quick, and does not necessitate high pressure, energy, temperature, or the use of highly toxic chemicals.

In this review, the authors describe mosquito-borne diseases and the cycle of transmission, current strategies developed against main vectors. Also, Green synthesized plant-based metallic nanoparticles as mosquito control strategy was focused.”

The paper is well structured and generally well written but too long.  

I recommend accepting this paper for publication in Insect pending major corrections.
Here are several suggestions for improvement:

Major comments.

1.     Provide briefly the main steps to produce Green synthesized plant-based metallic nanoparticles.

2.     Plant-derived nanoinsecticides can be subdivided into different forms such as plant-derived nanoparticles and nanoemulsion formulated using the essential oils derived from plant parts. In this review, though this paper focused on nanoparticles, I suggest authors to evoke the essential oils nanoemulsions which also remain as promising mosquito control. Can you add one section on this tool?

3.     Overall, the nanoparticles were found to be effective on the larval stage of mosquitoes. What is the mode of action of the nanoparticles? Have they toxic effect or they disrupt larvae growth?

4.     In the tables 1-8, provide the origin of type mosquitoes tested as well as the status of resistant (susceptible or resistant), that is permit to appreciate if this technology may be used an alternative tool against resistant strains. (please check Balbone et al, 2022a,b, c, Journal of medical entomology, scientific reports)

5.     In the table 2, Phenacoccus solenopsis is not mosquito, delete line related to adulticide effect on this insect.

6.     Figures 3, 4, 8 must to be improved, it is not clear.

7.     Line 657-660 on Wolbachia spp. section, improve this part on phenotypic effect of this bacteria. Reproductive manipulations exerted by Wolbachia on its hosts include male killing, feminization, parthenogenesis and cytoplasmic incompatibility (CI). In addition, Wolbachia have other effects. Please check Werren et al. 2008, Wolbachia: masters manipulators of invertebrate biology

8.     Regarding to the interesting role of nanoparticles in the control of most of vectors,  Do this methodology affects non targets organisms and mosquito environment where they were sprayed ?

9.     What is time of duration of nanoparticles in environment?

10.   In perspective, it will be need to investigate on the effect of nanoparticles on the toxicity of adults as well as on some life history traits.

Author Response

The paper is well structured and generally well written but too long.  

Based on your comment, the review has been reduced and restructured.

I recommend accepting this paper for publication in Insect pending major corrections.
Here are several suggestions for improvement:

Major comments.

  1. Provide briefly the main steps to produce Green synthesized plant-based metallic nanoparticles.

Based on your comment, the main steps to produce green metallic nanoparticles have been provided (section 4, lines 519-529).

  1. Plant-derived nanoinsecticides can be subdivided into different forms such as plant-derived nanoparticles and nanoemulsion formulated using the essential oils derived from plant parts. In this review, though this paper focused on nanoparticles, I suggest authors to evoke the essential oils nanoemulsions which also remain as promising mosquito control. Can you add one section on this tool?

The articles dealing with larvicides and repellent nanoemulsions based on essential oils derived from plant extracts are so numerous. We therefore felt that it deserves a separate article.

  1. Overall, the nanoparticles were found to be effective on the larval stage of mosquitoes. What is the mode of action of the nanoparticles? Have they toxic effect or they disrupt larvae growth?

The modes of action of green metallic nanoparticles towards larval and other satges of mosquitoes are not yet fully elucidated. Nevertheless, section 4.6. related to “possible mechanisms of action of mosquitocidal green metallic nanoparticles” has been added to evoke and discuss the current state of knowledge on the topic (lines 740-777).

  1. In the tables 1-8, provide the origin of type mosquitoes tested as well as the status of resistant (susceptible or resistant), that is permit to appreciate if this technology may be used an alternative tool against resistant strains. (please check Balbone et al, 2022a,b, c, Journal of medical entomology, scientific reports)

Based on your comment, we have gone through the literature reported in this review. It is resulted that, in general, the authors did not mention the origin and the status of resistance (susceptible or resistant) of tested mosquitoes.

To allow researchers to consider this weakness in the future, we have mentioned this in section 4.6., lines 688-690.

  1. In the table 2, Phenacoccus solenopsis is not mosquito, delete line related to adulticide effect on this insect.

Thanks. Data related to Phenacoccus solenopsis has been deleted.

  1. Figures 3, 4, 8 must to be improved, it is not clear.

In order to reduce Section 2, certain sentences and figures have been reduced or even deleted. This is the case of figures 3, 4 and 8 which have been removed.

  1. Line 657-660 on Wolbachia spp. section, improve this part on phenotypic effect of this bacteria. Reproductive manipulations exerted by Wolbachia on its hosts include male killing, feminization, parthenogenesis and cytoplasmic incompatibility (CI). In addition, Wolbachia have other effects. Please check Werren et al. 2008, Wolbachia: masters manipulators of invertebrate biology

Considering your comments, the subsection 3.2.7. (lines 329-345) of the revised version has been rewritten.

  1. Regarding to the interesting role of nanoparticles in the control of most of vectors,  do this methodology affects non targets organisms and mosquito environment where they were sprayed ?

Based on your question, the section 4.7. (lines 779-805) related to “non-target effects of mosquitocidal green metallic nanoparticles” has been added.

  1. What is time of duration of nanoparticles in environment?

Information related to the duration of nanoparticles in the environment is very limited. Thus, in section 4.7. (lines 804-805), we have noted this as a perspective to be taken into account in the future.

  1.  In perspective, it will be need to investigate on the effect of nanoparticles on the toxicity of adults as well as on some life history traits.

Your recommendation has been considered in section 4.6. related to the toxic effect of metallic nanoparticles on mosquitoes. A sentence has been added between lines 774 and 777.

Reviewer 2 Report

The title of this review suggests it will be focused on plant-based nanoparticles, but it spends most of its time reviewing mosquito-borne diseases.  This content has been extensively reviewed. The review needs major restructuring, as per my comments below. 

Abstract

Line 21: (grammar) mosquitoes act as vectors

Introduction lines 39-42: it’s not just growth of human populations but also climate change.  It would be better to remove these lines and just that increased globalization and mobility have led to an increase of highly invasive species reaching new habitats.

Line 47: mosquitoes don’t vector disease, they vector pathogens so please rephrase

Line 55: this is confusing – which novel arbovirus are the authors referring to, as multiple arboviruses have emerged?

Line 70:  should that be “The use and development OF nanomedicines… “

Line 75: typo/ proofreading

Line 77: grammar/ proofreading

Lines 103-104: why single out Lithuania?  It doesn’t really serve any purpose, so I would suggest deleting this.

Sections 2.1 through to 2.9 should be deleted.  This information is covered extensively in other reviews and really does not belong here.  The authors just need a short section listing the main mosquito-borne diseases, viral families and vectors, disease burdens and distribution. 

Figure 1: do the authors have permission to reproduce this figure?  In fact all of the figures appear to be from other papers so there will be copyright issues if there is no permission to reproduce them from the original publication and journal. 

Line 411: not all mosquitoes swarm, so this is incorrect

Lines 474-475: this line needs references. 

Line 517: specify which virus is being referred to here and add a full stop

Lines 533-536: it’s unclear how this paragraph relates to the rest of this section

Lines 537-539: these are vague statements so authors should consider rephrasing or deleting this paragraph

Lines 556-559: this sentence is very difficult to understand.

Line 560: this statement deserves to have more information; how do these bioinsecticides work?

Line 587: replace the word employment with use so it reads ‘their use…’

Lines 624-635: consider rephrasing this sentence as it suggests insects and bacteria cause cancer in humans, which is incorrect

4.2.7 heading: Wolbachia spp. should be in italics

4.2.8 heading: Asaia spp. should be in italics

Line 704: what is meant here by ‘unprecedented’?

Section 4.3.2: the plural of mosquito is not mosquitos but mosquitoes

Section 4.3.3 seems out of place, because it’s not strictly insect vector control unless the authors are referring to endectocides.  If it’s endectocides, then the section needs to be expanded to cover this topic properly.

Section 4.5 repeats a lot of what is said previously.  The authors could leave this section in after deletion of sections 2.1 to 2.9

Line 875: I think the authors have forgotten to put in a number when referring to the plant families

Line 926: the authors need to define what green metal nanoparticles are

Line 977: Replace ‘in our mind’ with ‘To the best of our knowledge,…’

Line 985: test should be plural, so ‘tests’

Tables 1-8 would be much better presented as Supplementary tables; the authors could present a summary table instead

Line 1048: replace ‘inferior’ with ‘lower’

Figure 10: is this an exhaustive review of the articles? How were these papers identified for this meta-analysis? How do the authors know they have captured all the relevant articles?  

Author Response

  1. The title of this review suggests it will be focused on plant-based nanoparticles, but it spends most of its time reviewing mosquito-borne diseases.  This content has been extensively reviewed. The review needs major restructuring, as per my comments below. 

Based on your comment, the review has been restructured. We have significantly reduced the section related to "Mosquito-borne diseases".

  1. Line 21: (grammar) mosquitoes act as vectors

The mistake has been corrected.

  1. Introduction lines 39-42: it’s not just growth of human populations but also climate change.  It would be better to remove these lines and just that increased globalization and mobility have led to an increase of highly invasive species reaching new habitats.

The aforementioned lines have been restructured and information related to climate change, increased globalization and mobility added (lines 51-52).

  1. Line 47: mosquitoes don’t vector disease, they vector pathogens so please rephrase

This sentence has been rephrased (along with a similar one in the abstract).

  1. Line 55: this is confusing – which novel arbovirus are the authors referring to, as multiple arboviruses have emerged?

The authors were referring to all the arboviruses that have emerged, not only one. This sentence has been rephrased (line 65).

  1. Line 70:  should that be “The use and development OF nanomedicines… “

Avowing an error, this sentence has been rephrased

  1. Line 75: typo/ proofreading

The correction has been made.

  1. Line 77: grammar/ proofreading

Corrected: the word “an” has been deleted (line 89).

  1. Lines 103-104: why single out Lithuania?  It doesn’t really serve any purpose, so I would suggest deleting this.

The sentence has been deleted.

  1. Sections 2.1 through to 2.9 should be deleted.  This information is covered extensively in other reviews and really does not belong here.  The authors just need a short section listing the main mosquito-borne diseases, viral families and vectors, disease burdens and distribution. 

According to your comment, these sections have been rewritten and reduced. Now, the new section briefly includes only the main mosquito-borne diseases, their vectors as well as their disease burdens and distribution.

  1. Figure 1: do the authors have permission to reproduce this figure?  In fact all of the figures appear to be from other papers so there will be copyright issues if there is no permission to reproduce them from the original publication and journal. 

Since we have reduced the sections 2.1 through to 2.9., we have removed all the figures except those designed by ourselves.

  1. Line 411: not all mosquitoes swarm, so this is incorrect

This line has been deleted with its entire section deemed unnecessary based on the comments of the reviewers.

  1. Lines 474-475: this line needs references. 

In order to reduce the length of the review, the section related to “control strategies for mosquito-borne diseases” has been restructured. Reference has been added in the revised version (line 193-195).

  1. Line 517: specify which virus is being referred to here and add a full stop

In order to reduce the length of the review, the section related to “control strategies for mosquito-borne diseases” have been restructured. This line has been rephrased in the revised version (lines 235-236).

  1. Lines 533-536: it’s unclear how this paragraph relates to the rest of this section

In order to be more understandable by the readers, this paragraph has been rephrased (lines 250-253).

  1. Lines 537-539: these are vague statements so authors should consider rephrasing or deleting this paragraph

This paragraph has been deleted.

  1. Lines 556-559: this sentence is very difficult to understand.

In order to reduce the length of the review, the section related to “control strategies for mosquito-borne diseases” have been restructured. The subsection related to “genetic modification” has been completely reduced and rephrased (lines 260-266). It becomes more understandable.

  1. Line 560: this statement deserves to have more information; how do these bioinsecticides work?

Based on your comment that the manuscript is too long, the section related to “control strategies for mosquito-borne diseases” have been reduced. Hence, information/sentence related to “RNA interference –based bioinsecticides” has been deleted.

  1. Line 587: replace the word employment with use so it reads ‘their use…’

This line has been rephrased (line 286-288).

  1. Lines 624-635: consider rephrasing this sentence as it suggests insects and bacteria cause cancer in humans, which is incorrect

This sentence has been rephrased as follow: “Some of these proteins which are toxic to a variety of insect orders and nematodes, are known to cause cancer in humans if mishandled” (line 309-311).

  1. 4.2.7 heading: Wolbachia spp. should be in italics

The correction has been made.

  1. 4.2.8 heading: Asaia spp. should be in italics

The correction has been made.

  1. Line 704: what is meant here by ‘unprecedented’?

This word has been deleted and the sentence rephrased.

  1. Section 4.3.2: the plural of mosquito is not mosquitos but mosquitoes

The error has been corrected in this section and throughout the manuscript.

  1. Section 4.3.3 seems out of place, because it’s not strictly insect vector control unless the authors are referring to endectocides.  If it’s endectocides, then the section needs to be expanded to cover this topic properly.

This section has been deleted.

  1. Section 4.5 repeats a lot of what is said previously.  The authors could leave this section in after deletion of sections 2.1 to 2.9

To avoid redundancies and reduce the lengthy of the manuscript, some information related to this section has been summarized and inserted in the section 2 related to “mosquito-borne diseases”.

  1. Line 875: I think the authors have forgotten to put in a number when referring to the plant families

Exact. The word “several” has been added so as the sentence makes sense (line 455).

  1. Line 926: the authors need to define what green metal nanoparticles are

A definition has been added (lines 510-515 and 519-529).

  1. Line 977: Replace ‘in our mind’ with ‘To the best of our knowledge,…’

The correction has been made as suggested by the reviewer (line 579).

  1. Line 985: test should be plural, so ‘tests’

The correction has been made (line 588).

  1. Tables 1-8 would be much better presented as Supplementary tables; the authors could present a summary table instead

As advised, the tables have been grouped into two. The first named Table 1 contains only 30 references and is presented in the manuscript. The second one is presented as a supplemental table (Table S1).

  1. Line 1048: replace ‘inferior’ with ‘lower’

The correction has been made (line 663).

  1. Figure 10: is this an exhaustive review of the articles? How were these papers identified for this meta-analysis? How do the authors know they have captured all the relevant articles?

Figures 9 and 10 (that become Figure 1 and 2 in the revised manuscript) have been drawn by us based on all the data listed in Table 1 and S1. The latter include all the relevant articles indexed in Pubmed, Sciencedirect and Google scholar. The information has been given in the revised version (section 4.3., lines 681-682).

Reviewer 3 Report

The draft of article number 2095625 submitted to the Insects, MDPI, entitled “Mosquito-borne diseases and their control strategies: an overview focused on green-synthesized plant-based metallic nanoparticles” carried text that needs thorough revision for the improvement of the draft. The manuscript is without focused objectives. In addition, the introduction, tables, and conclusion portions are not precisely written and the figures are without mentioning the source data. The publication of this study needs revision. Some suggested corrections for example are in the comments portion to revise and improve the manuscript. There are many sentences throughout the manuscript which is hard to understand. Please find suggested corrections, reference writing, journal-style format, author’s instructions, use of abbreviations, and missing information for revision. 

Mosquito-borne diseases and their control strategies: an overview focused on green-synthesized plant-based metallic nanoparticles

Please rephrase the title of the paper

This review paper is not focused on the green-synthesized plant-based metallic nanoparticles” please rephrase/revise the objectives

(Dengue virus, Rift Valley fever virus, Tick-borne encephalitis virus, West Nile virus, Yellow fever virus, etc.)” remove the word virus as this is used many times

The research paper is too descriptive without focusing on the objective of the paper.

The paper is about the Aedes species and or whole mosquito fauna” the objectives of the paper may be rephrased” please also consult the author’s instructions

The research paper is a review of mosquito-borne diseases and their control approaches rather than nanoparticles control

The objectives of the paper are confusing: please recheck the objectives and corrections may be made according to the journal’s author instructions before submission

The introduction of the paper may be rechecked and written briefly/concisely

Please recheck if the Mosquito biology and life cycle are necessary to include in the paper” please be concise in writing review

4.2. Biological control” portion is too little and without references

Chemical control strategies are also not fully covered

Tables 1-7: some tables may be included in the supplementary tables by mentioning in main the text

Figure 9-10: Please mention the years and source of the data included

Please also give the references/ sources of the figures if included in published materials

Conclusion and perspectives are not concise

References are too many” please include only the most relevant references with the text

Please write the references in the text and in the reference portion according to the author's instructions and journal style/formatting. Please double-check for inconsistencies in Journal style/formatting/ authors instructions, double spaces, spellings of the words, English vocabulary, missing italics, scientific names, excessive/missing information, etc.

Author Response

The draft of article number 2095625 submitted to the Insects, MDPI, entitled “Mosquito-borne diseases and their control strategies: an overview focused on green-synthesized plant-based metallic nanoparticles” carried text that needs thorough revision for the improvement of the draft.

  1. The manuscript is without focused objectives.

Based on your comment, the manuscript has been revised to focus on the objectives.

  1. In addition, the introduction, tables, and conclusion portions are not precisely written and the figures are without mentioning the source data.
  2. a) Introduction and conclusion have been rephrased.

b) As advised, the tables are now more precise since they have been grouped into two. The first named Table 1 contains only 30 references and is presented in the manuscript. The second one is presented as a supplemental table (Table S1).

  1. c) Since we have reduced the sections 2.1 through to 2.9., we have removed all the figures except those designed by ourselves i.e. Figures 9 and 10 (that become Figure 1 and 2 in the revised manuscript).
  2. The publication of this study needs revision. Some suggested corrections for example are in the comments portion to revise and improve the manuscript. There are many sentences throughout the manuscript which is hard to understand. Please find suggested corrections, reference writing, journal-style format, author’s instructions, use of abbreviations, and missing information for revision. 

The revision has been made. The revised manuscript has been much improved by following your recommendations. Some sentences have been revised for easier understanding: the meaning of the abbreviations has been given, reference writing checked, etc.

  1. Mosquito-borne diseases and their control strategies: an overview focused on green-synthesized plant-based metallic nanoparticles. Please rephrase the title of the paper.

Since the manuscript has been restructured and revised as well as many sentences rephrased to achieve the objectives of the review, we think that the title can be maintained.

  1. This review paper is not focused on the green-synthesized plant-based metallic nanoparticles” please rephrase/revise the objectives

Based on your comment, the paper has been revised and restructured as well as many sentences deleted or rephrased to focus on the green-synthesized plant-based metallic nanoparticles.

  1. (Dengue virus, Rift Valley fever virus, Tick-borne encephalitis virus, West Nile virus, Yellow fever virus, etc.)” remove the word virus as this is used many times

The correction has been made in the abstract.

  1. The research paper is too descriptive without focusing on the objective of the paper.

Based on your comment, the paper has been revised and restructured as well as many sentences deleted or rephrased to focus on the green-synthesized plant-based metallic nanoparticles.

  1. The paper is about the Aedes species and or whole mosquito fauna” the objectives of the paper may be rephrased” please also consult the author’s instructions

The paper is about mosquito-borne diseases. However, the main mosquito vectors belong to the three genera (Aedes, Culex and Anopheles), as mentioned in the introduction

  1. The research paper is a review of mosquito-borne diseases and their control approaches rather than nanoparticles control. The objectives of the paper are confusing: please recheck the objectives and corrections may be made according to the journal’s author instructions before submission

Based on your comment, the paper has been revised and restructured as well as many sentences deleted or rephrased to focus on the green-synthesized plant-based metallic nanoparticles.

  1. The introduction of the paper may be rechecked and written briefly/concisely

Based on your comment, the introduction has been rewritten.

  1. Please recheck if the Mosquito biology and life cycle are necessary to include in the paper” please be concise in writing review

Indeed, this section is unnecessary so it has been and deleted.

  1. 4.2. Biological control” portion is too little and without references. Chemical control strategies are also not fully covered.

To focus on the objectives of the review, these sections have been rephrased.

  1. Tables 1-7: some tables may be included in the supplementary tables by mentioning in main the text

As advised, the tables are now more precise since they have been grouped into two. The first named Table 1 contains only 30 references and is presented in the manuscript. The second one is presented as a supplemental table (Table S1).

  1. Figure 9-10: Please mention the years and source of the data included

Figures 9 and 10 (that become Figure 1 and 2 in the revised manuscript) have been drawn by ourselves.

  1. Please also give the references/ sources of the figures if included in published materials

Since we have reduced the sections 2.1 through to 2.9., we have removed all the figures except those designed by ourselves i.e. Figures 9 and 10 (that become Figure 1 and 2 in the revised manuscript).

  1. Conclusion and perspectives are not concise

The conclusion has been rephrased to be more concise

  1. References are too many” please include only the most relevant references with the text

The manuscript includes only the most relevant references. Some of them have been deleted in the revised version.

  1. Please write the references in the text and in the reference portion according to the author's instructions and journal style/formatting.

The correction has been made where needed.

  1. Please double-check for inconsistencies in Journal style/formatting/ authors instructions, double spaces, spellings of the words, English vocabulary, missing italics, scientific names, excessive/missing information, etc.

The correction has been made where needed.

Round 2

Reviewer 1 Report

insects-2095625

TitleMosquito-borne diseases and their control strategies: an overview focused on green-synthesized plant-based metallic nanoparticles

Comments to the authors

Most of the comments were reviewed and adjusted. However, one comment is still needing  for clarification.

In table I, for instance, it is necessary to clarify the origin of mosquitoes in which the various bioassays have done. It is wild populations that have most time under insecticide pressure or laboratory strains (susceptible strains). I suggest to insert one column related to “origin of mosquitoes”.

I accept it  after minor revision

Author Response

In table I, for instance, it is necessary to clarify the origin of mosquitoes in which the various bioassays have done. It is wild populations that have most time under insecticide pressure or laboratory strains (susceptible strains). I suggest to insert one column related to “origin of mosquitoes”.

I accept it  after minor revision

Based on your comment, the origin of mosquitoes has been inserted in Table 1 and S1.

Reviewer 2 Report

The paper is significantly improved but please see the below suggestions.  

Summary is fine but carefully edit for correct grammar

Line 32: insert the word agents after ovicidal, larvicidal, pupicidal and adulticidal…    

Line 37: spread instead of spreading

Line 46: is this accurate to state as an objective?  This review should not be about different mosquito control strategies, which is an immense literature, but about plant-mediated mosquitocidals

Line 50: remove the word ‘improved’

Line 56: can you please be consistent in the use of capitals for virus names? Earlier in the paper all were capitalised, yet now they are not.  The same inconsistencies crop up throughout the paper (e.g. lines 88, 336-337)

Line 84: it’s unclear whether the strengths and weaknesses are of the control approaches or the nanoparticles.  Please rephrase to clarify.

Line 93: replace the word into with in

Line 106: it’s unclear what is meant by transmission by birds, since we now the virus does infect birds in all areas where it’s found, not just Africa and Europe.

Lines 115-116: grammar; pathogens are..

Line 113: yellow fever virus – word fever missing

Line 121: what is meant by the intermediate cycle?  It hasn’t been defined in the text.

Line 122: Chikungunya disease is cause by the alphavirus chikungunya virus

Line 135: remove the word ‘also’

Line 161: once a virus is abbreviated, the authors should use the abbreviation throughout.

Line 178/179 – grammar

Line 192: typo

Line 196: grammar

Line 199: what is meant by ‘low-performing’?  The authors have just said that it’s actually high performing through massive reductions in malaria.  Do they mean low cost but high performing?

Line 200 – scaled, not scalled.

Line 209: incorrect spelling of prevalence

Line 229: transovarial transmission of these viruses is very rare and not a major cause of infections with these viruses

Line 262: add ‘in some instances’ at the end of that sentence, as SIT has been successful in other occasions.

Lines 288: the phrase ‘pathogen protozoan parasite’ does not make sense; do the authors mean a parasite that is pathogenic??

Section 3.2.6 – are there any off-target effects to non-mosquito species from IGRs?

Line 329:  this should be cytoplasmic incompatibility, not compatibility

Lines 332-333: this sentence does not make sense as it suggests Ae. polynesiensis is being used to eradicate Ae. albopictus

Line 334: potential resistance to what?  Virus counter-evolution or insecticides? Please clarify and state.

Line 354: proofreading required

Section 3.2.10 – how would these be used? On clothes or as lotions/ creams? This should be mentioned for the reader.

Line 417: mosquitoes are endemic in many places so I would suggest rephrasing this to emphasise pathogen-transmitting genera

Line 457: replace the word from with the word using. 

Line 488 -489: this sentence does not logically fit with the rest of the paragraph. How is home modification related to plants? 

Line 576: no need for an apostrophe after mosquitoes; should be ‘mosquito egg rafts…’

Line 586: replace with ‘eggs that were no more than 4 hours old’

Line 591: typo

Line 673: shouldn’t this just be Figure 2 in brackets?  It’s best to talk about figures separately, otherwise it becomes confusing as to which figure shows what.  

Figure 2 needs a bit more elaboration as it's not always clear from the text what is being shown there.  

Line 677: susceptibility to what? Other insecticides? Please clarify. 

Figure 1 Legend: The figure needs a unifying title, not just the different sections.  Section C, the names of mosquito spp. should be given in italics.

Line 776: remove italics from word mosquito

Section 4.7.  Is there any evidence of off-target effects on other insects, aside from Toxorhynchites

Author Response

Comments and Suggestions for Authors

The paper is significantly improved but please see the below suggestions.  

1) Summary is fine but carefully edit for correct grammar

The grammatical errors have been corrected (lines 24, 25 and 27).

2) Line 32: insert the word agents after ovicidal, larvicidal, pupicidal and adulticidal…    

The word has been added (line 32)

3) Line 37: spread instead of spreading

The error has been corrected (line 38)

4) Line 46: is this accurate to state as an objective?  This review should not be about different mosquito control strategies, which is an immense literature, but about plant-mediated mosquitocidals

Based on your comment and considering the title of our review, the sentence has been rephrased by adding “in particular” (line 49).

5) Line 50: remove the word ‘improved’

The word has been removed (line 52)

6) Line 56: can you please be consistent in the use of capitals for virus names? Earlier in the paper all were capitalised, yet now they are not.  The same inconsistencies crop up throughout the paper (e.g. lines 88, 336-337)

The inconsistencies have been corrected (lines 58, 92, 117, 236 and 238).

7) Line 84: it’s unclear whether the strengths and weaknesses are of the control approaches or the nanoparticles.  Please rephrase to clarify.

The sentence has been rephrased (line 87).

8) Line 93: replace the word into with in

The word has been replaced (line 96).

9) Line 106: it’s unclear what is meant by transmission by birds, since we now the virus does infect birds in all areas where it’s found, not just Africa and Europe.

The sentence has been rephrased (lines 108-109).

10) Lines 115-116: grammar; pathogens are..

The correction has been done (line 120).

11) Line 113: yellow fever virus – word fever missing

The missing word has been added (line 117)

12) Line 121: what is meant by the intermediate cycle?  It hasn’t been defined in the text.

The explanation has been given (lines 125-126).

13) Line 122: Chikungunya disease is cause by the alphavirus chikungunya virus

The sentence has been rephrased (line 127).

14) Line 135: remove the word ‘also’

The word has been removed (line 141)

15) Line 161: once a virus is abbreviated, the authors should use the abbreviation throughout.

The sentence has been rephrased (line 167).

16) Line 178/179 – grammar

The errors have been corrected (lines 186-187).

17) Line 192: typo

The errors have been corrected (line 199).

18) Line 196: grammar

The grammatical error has been corrected (line 203).

19) Line 199: what is meant by ‘low-performing’?  The authors have just said that it’s actually high performing through massive reductions in malaria.  Do they mean low cost but high performing?

The sentence has been rephrased (line 207).

20) Line 200 – scaled, not scalled.

The correction has been done (line 207)

21) Line 209: incorrect spelling of prevalence

The correction has been done (line 216).

22) Line 229: transovarial transmission of these viruses is very rare and not a major cause of infections with these viruses

The sentence has been rephrased (lines 236-237).

23) Line 262: add ‘in some instances’ at the end of that sentence, as SIT has been successful in other occasions.

“in some instances” has been added (line 268).

24) Lines 288: the phrase ‘pathogen protozoan parasite’ does not make sense; do the authors mean a parasite that is pathogenic??

The sentence has been rephrased (line 297).

25) Section 3.2.6 – are there any off-target effects to non-mosquito species from IGRs?

To the best of knowledge, IGRs are selective as mentioned in line 327. May be others studies are required to confirm their selectivity.

26) Line 329:  this should be cytoplasmic incompatibility, not compatibility

The correction has been done (line 337)

27) Lines 332-333: this sentence does not make sense as it suggests Ae. polynesiensis is being used to eradicate Ae. albopictus

The sentence has been rephrased (line 342)

28) Line 334: potential resistance to what?  Virus counter-evolution or insecticides? Please clarify and state.

The sentence has been rephrased (lines 342-345).

29) Line 354: proofreading required

The correction has been done (line 364).

30) Section 3.2.10 – how would these be used? On clothes or as lotions/ creams? This should be mentioned for the reader.

Based on your comment, a sentence has been added (lines 391-392).

31) Line 417: mosquitoes are endemic in many places so I would suggest rephrasing this to emphasise pathogen-transmitting genera

The sentence has been rephrased (line 430).

32) Line 457: replace the word from with the word using. 

The word “from” has replaced (line 470).

33) Line 488 -489: this sentence does not logically fit with the rest of the paragraph. How is home modification related to plants? 

The sentences have been rephrased (lines 503-507).

34) Line 576: no need for an apostrophe after mosquitoes; should be ‘mosquito egg rafts…’

The correction has been made (line 595).

35) Line 586: replace with ‘eggs that were no more than 4 hours old’

The sentence has been rephrased (line 605).

36) Line 591: typo

The typo has been corrected (line 610).

37) Line 673: shouldn’t this just be Figure 2 in brackets?  It’s best to talk about figures separately, otherwise it becomes confusing as to which figure shows what.  

The corrections have been made (lines 691 and 694).

38) Figure 2 needs a bit more elaboration as it's not always clear from the text what is being shown there.  

For more clarification, a unifying title has been added (lines 700 and 708).

39) Line 677: susceptibility to what? Other insecticides? Please clarify. 

Based on the comment from another reviewer, the sentence has been rephrased (lines 695-696).

40) Figure 1 Legend: The figure needs a unifying title, not just the different sections. 

A unifying title has been added (lines 700 and 708)

41) Section C, the names of mosquito spp. should be given in italics

Unfortunately, the software does not give the possibility to put in italics.

42) Line 776: remove italics from word mosquito

Italic has been removed (line 794).

43) Section 4.7.  Is there any evidence of off-target effects on other insects, aside from Toxorhynchites

Based on your comment, “no obvious” has been added in the sentence (line 794).

Reviewer 3 Report

The revised draft of article number 2095625 submitted to the Insects, MDPI, entitled “Mosquito-borne diseases and their control strategies: an overview focused on green-synthesized plant-based metallic nanoparticles” carried text that needs some revision for the improvement of the draft. The publication of this study needs some revision. Some suggested changes for example are in the comments portion to revise the manuscript.

Thus, scientists are searching ……forms of insecticides.” Please rephrase the sentence by adding researchers and also academicians

The literature reported several studies…..given in Tables 1 and S1.” Please combine table 1 with supplementary table S1 in the supplementary portion.

Figure 1 and Figure 2 may be shifted to supplementary figures

Please double-check for inconsistencies in journal style/formatting/ Reference writings/authors' instructions, double spaces, spellings of the words, English vocabulary, missing italics, scientific names, excessive/missing information, etc.

Author Response

Comments and Suggestions for Authors

The revised draft of article number 2095625 submitted to the Insects, MDPI, entitled “Mosquito-borne diseases and their control strategies: an overview focused on green-synthesized plant-based metallic nanoparticles” carried text that needs some revision for the improvement of the draft. The publication of this study needs some revision. Some suggested changes for example are in the comments portion to revise the manuscript.

1) Thus, scientists are searching ……forms of insecticides.” Please rephrase the sentence by adding researchers and also academicians

The sentence has been rephrased (line 27).

2) The literature reported several studies…..given in Tables 1 and S1.” Please combine table 1 with supplementary table S1 in the supplementary portion.

We certainly agree with the reviewer. However, to avoid redundancy and considering comments from other reviewers, we feel that it would be best to keep the tables as they are.

3) Figure 1 and Figure 2 may be shifted to supplementary figures.

Considering that the data presented in the figures are crucial for the readers and based on the comments from other reviewers, we feel that it would be best to keep the figures as they are in the body of the manuscript.

4) Please double-check for inconsistencies in journal style/formatting/ Reference writings/authors' instructions, double spaces, spellings of the words, English vocabulary, missing italics, scientific names, excessive/missing information, etc.

Spellings of the words, English vocabulary, missing italics, scientific names, excessive/missing information has been corrected or added (lines 24, 25, 32, 38, 52, 96, 120, 141, 186, 187, 203, 207, 216, 231, 364, 389, 470, 610, 794, etc.).

Regarding the double spaces and policy, we believe that the MDPI team will take care of them when designing the final proof.